# Compositional Transformers for Scene Generation

**Drew A. Hudson**
Department of Computer Science
Stanford University
dorarad@cs.stanford.edu

**C. Lawrence Zitnick**
Facebook AI Research
Facebook, Inc.
zitnick@fb.com

## Abstract

We introduce the GANformer2 model, an iterative object-oriented transformer, explored for the task of generative modeling. The network incorporates strong and explicit structural priors, to reflect the compositional nature of visual scenes, and synthesizes images through a sequential process. It operates in two stages: a fast and lightweight *planning* phase, where we draft a high-level scene layout, followed by an attention-based *execution* phase, where the layout is being refined, evolving into a rich and detailed picture. Our model moves away from conventional black-box GAN architectures that feature a flat and monolithic latent space towards a transparent design that encourages efficiency, controllability and interpretability. We demonstrate GANformer2's strengths and qualities through a careful evaluation over a range of datasets, from multi-object CLEVR scenes to the challenging COCO images, showing it successfully achieves state-of-the-art performance in terms of visual quality, diversity and consistency. Further experiments demonstrate the model's disentanglement and provide a deeper insight into its generative process, as it proceeds step-by-step from a rough initial sketch, to a detailed layout that accounts for objects' depths and dependencies, and up to the final high-resolution depiction of vibrant and intricate real-world scenes. See https://github.com/dorarad/gansformer for model implementation.

## 1  Introduction

Drawing, the practice behind human visual and artistic expression, can essentially be defined as an iterative process. It starts from an initial outline, with just a few strokes that specify the spatial arrangement and overall layout, and is then gradually refined and embellished with color, depth and richness, until a vivid picture eventually emerges. These initial schematic sketches can serve as an abstract scene representation that is very concise, yet highly expressive: several lines are enough to delineate three dimensional structures, account for perspective and proportion, convey shapes and geometry, and even imply semantic information [52, 63, 67]. A large body of research in psychology highlights the importance of sketching in stimulating creative discovery [40, 58], fostering cognitive development [27], and facilitating problem solving [26, 44]. In fact, a large variety of generative and RL tasks either in the visual [5], textual [22, 78] or symbolic modalities [2], can benefit from the same strategy – prepare a high-level plan first, and then carry out the details.

At the heart of this hierarchical strategy stands the principle of compositionality – where the meaning of the whole can be derived from those of its constituents. Indeed, the world around us is highly structured. Our environment consists of a varied collection of objects, tightly interconnected through dependencies of all sorts: from close-by to long-range, and from physical and dynamic to abstract and semantic [10, 66]. As pointed out by prior literature [24, 51, 55, 57], compositional representations are pivotal to human intelligence, supporting our capabilities of reasoning [32], planning [53], learning [54] and imagination [6]. Realizing this principle, and explicitly capturing the objects and elements composing the scene, is thereby a desirable goal, that can make the generative process more explainable, controllable, versatile and efficient.

35th Conference on Neural Information Processing Systems (NeurIPS 2021).

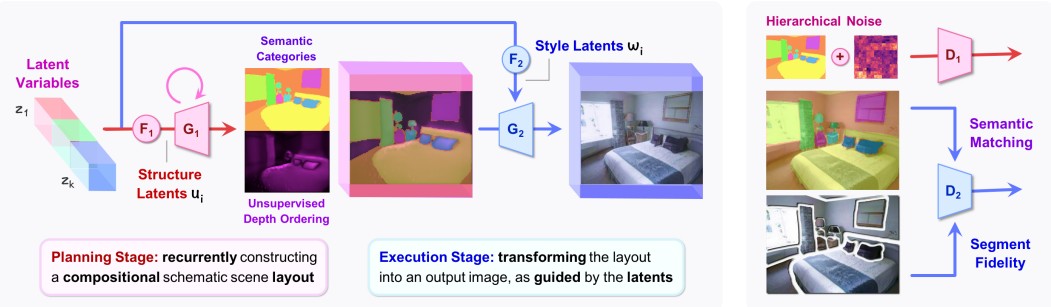

Figure 1: **The GANformer2 model** has a set of latent variables that represent the objects and entities within the generated scene. It proceeds through two stages, **planning** and **execution**: first iteratively drafting a high-level layout of the scene's spatial structure, and then refining and translating it into a photo-realistic image. Each latent corresponds to a different layout's segment, controlling its structure during the planning, and its style during the execution. On the discriminator side, we introduce new structural losses for compositional scene generation.

Yet, the vast majority of generative models do not directly reflect this compositionality so intrinsic to visual scenes. Rather, most networks, and GANs in particular, aim to generate the entire scene all at once from a single monolithic latent space, through a largely opaque transformation. Consequently, while they excel in producing stunning, strikingly-realistic portrayals of faces, sole centralized objects, or natural scenery, they still struggle to faithfully mimic richer or densely-packed scenes, as those common to everyday contexts, falling short of mastering their nuances and intricacies [5, 50, 76]. Likewise, while there is some evidence for property disentanglement within their latent space to independent axes of variation at the global scale [28, 36, 64, 71], most existing GANs fail to provide controllability at the level of the individual object or local region. Understanding of the mechanisms by which these models give rise to an output image, and the transparency of their synthesis process, thereby remain rather limited.

Motivated to alleviate these shortcomings and make generative modeling more compositional, interpretable and controllable, we propose GANformer2, a structured object-oriented transformer that decouples the visual synthesis task into two stages: **planning** and **execution**. We begin by sampling a set of latent variables, corresponding to the objects and entities that compose the scene. Then, at the planning stage, we transform the latents into a schematic layout – a scene representation that depicts the object segments, reflecting their shapes, positions, semantic classes and relative ordering. Its construction occurs in an iterative manner, to capture the dependencies and interactions between different objects. Next, at the execution stage, the layout is translated into the final image, with the latents cooperatively guiding the content and style of their respective segments through bipartite attention [33]. This approach marks a shift towards a more natural process of image generation, in which objects and the relations among them are explicitly modeled, while the initial sketch is successively tweaked and refined to ultimately produce a rich photo-realistic scene.

We study the model's behavior and performance through an extensive set of experiments, where it attains state-of-the-art results in both conditional and unconditional synthesis, as measured along metrics of fidelity, diversity and semantic consistency. GANformer2 reaches high versatility, demonstrated through multiple datasets of challenging simulated and real-world scenes. Further analysis illustrates its capacity to manipulate chosen objects and properties in an independent manner, achieving both spatial disentanglement and separation between structure and style. We notably observe amodal completion of occluded objects, likely driven by the sequential nature of the computation. Meanwhile, inspection of the produced layouts and their components shed more light upon the synthesis of each resulting scene, substantiating the model's transparency and explainability. Overall, by integrating strong compositional structure into the model, we can move in the right direction towards multiple favorable properties: increasing robustness for rich and diverse visual domains, enhancing controllability over individual objects, and improving the generative process's interpretability.

## 2 Related Work

Over the last years, the field of visual sythesis has witnessed astonishing progress, driven in particular by the emergence of Generative Adversarial Networks [23]. Tremendous strides have been made in a wide variety of tasks, from image [11] to video generation [7], to super-resolution [45], style transfer [15] and translation [35]. Meanwhile, a growing body of literature explored object discovery and

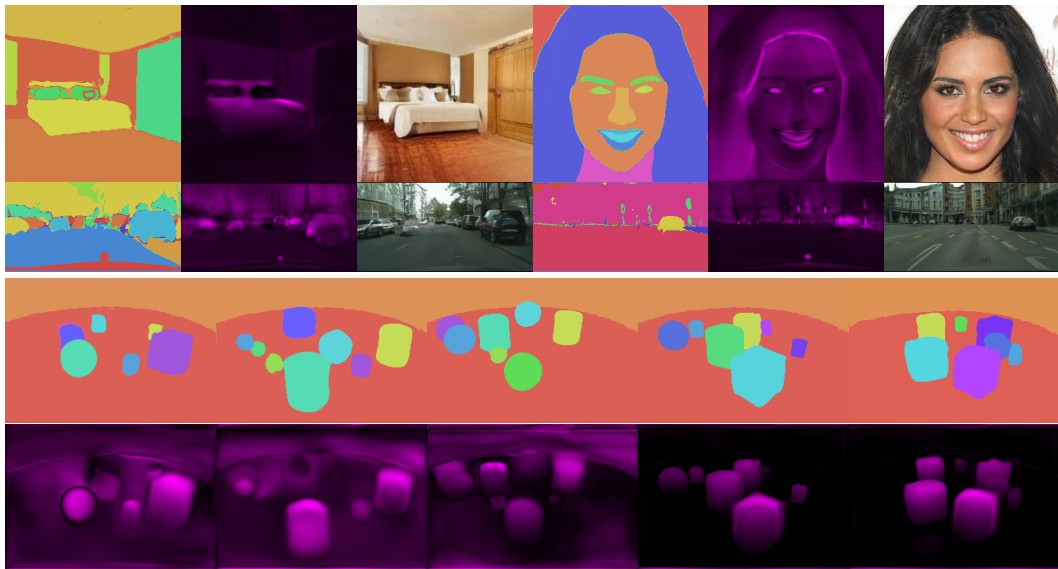

Figure 2: **Layout and image generation.** The GANformer2 model generates each image by first creating a layout that specifies its outline and object composition, which is then translated into the final image. While doing so, the model produces depth maps of the object segments which are used to determine their relative ordering, and notably are implicitly derived by the model from the layouts in a fully unsupervised manner.

sequential scene decomposition, albeit in the context of variational inference [1, 16, 19, 49]. DRAW [25] uses recurrent attention to encode and decode MNIST digits, while AIR [20], MONet [12] and IODINE [24] reconstruct simple simulated scenes by iteratively factorizing them into semantically meaningful segments. We draw inspiration from this pioneering line of works and marry the merits of its strong structure with the scalability and fidelity obtained by GANs, yielding a compositional model that successfully creates intricate real-world scenes, while offering a more transparent and controllable synthesis along the way, conceptually meant to better resemble the causal generative process and serve as a more natural emulation of how a human might approach the creative task.

To this end, our model leverages the GANformer architecture we introduced earlier this year [33] – a *bipartite transformer* that uses soft-attention to cooperetaviely translate a set of latents into a generated image [33, 70]. In GANformer2, we take this idea a step further, and show how we can use the bipartite transformer as a backbone for a more structured process, that enables not only an implicit, but also an *explicit modeling* of the objects and constituents composing the image's scene. Our **iterative** design endows the GANformer2 model with the flexibility to consider the correlations and interactions not only among latents but between synthesized scene elements directly, circumventing the need for partial independence assumptions made by earlier works [18, 56, 69], and thereby managing to scale to natural real-world scenes. Concurrently, the explicit **object-oriented** structure enhances the model's compositionallity so to outperform past coarse-to-fine cascade [80], layered [34], and hierarchical approaches [5], which, like us, employ a two-stage layout-based procedure, but contrary to our work, do so in a non-compositional fashion, and indeed were mostly explored within limited domains, such as indoor NYUv2 [73], faces [41], pizzas [59], or bird photographs [77].

Our model links to research on conditional generative modeling [72, 85, 87], including semantic image synthesis and text-to-image generation. The seminal Pix2Pix [35] and SPADE [60] works respectively adopt either a U-Net [62] or spatially-adaptive normalization [61] to map a given input layout into an output high-resolution image. However, beyond their notable weakness of being conditional and thus unable to create scenes from scratch, their reliance on both static class embeddings for feature modulation, along with perceptual or feature-matching reconstruction losses, render their sample variation rather low [68, 86]. In our work, we mitigate this issue by proposing new purely generative structural loss functions: *Semantic Matching* and *Segment Fidelity*, which maintain the one-to-many relation that holds between layouts and images, and consequently, as demonstrated in section 3.3, significantly enhance the output diversity.

While GANformer2 is an unconditional model, it shares some high-level ideas with conditional text-to-image [30, 31] and scene-to-image [4, 38] techniques, which use semantic layouts as a

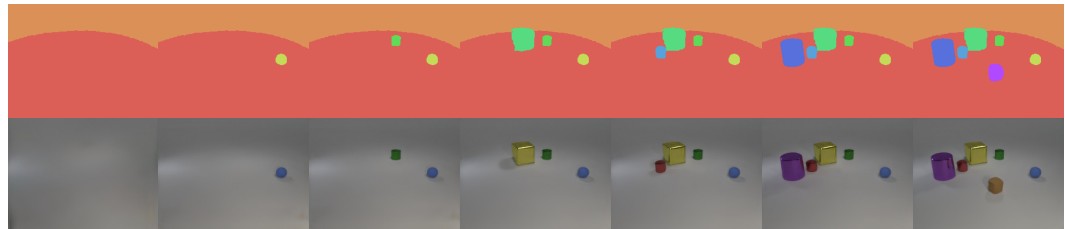

Figure 3: **Recurrent scene generation.** GANformer2 creates the layout sequentially, segment-by-segment, to capture the scene's compositionality, effectively allowing us to add or remove objects from the resulting images.

bottleneck intermediate representation. But whereas these works focus on heavily-engineered pipelines consisting of multiple pre- and post-processing stages and different sources of supervision, including textual descriptions, scene graphs, bounding boxes, segmentation masks and images, we propose a simpler, more elegant design, with a streamlined architecture and fewer supervision signals – relying on images and auto-predicted layouts only, to widen its applicability across domains. Finally, our approach relates to prior works on visual editing [8, 9, 46, 84] and image compositing [3, 13, 47, 79], but while these methods modify an existing source image, GANformer2 stands out being capable of creating structured manipulatable scenes from scratch.

## 3 The GANformer2 model

The GANformer2 model is a compositional object-oriented transformer for visual scene generation. It features a set of latent variables corresponding to the objects and entities composing the scene, and proceeds through two synthesis stages: (1) a sequential **planning stage**, where a scene layout – a collection of interacting object segments, is being created (section **3.2**), followed by (2) a parallel **execution stage**, where the layout is being refined and translated into an output photo-realistic image (section **3.3**). Both stages are implemented using customized versions of the GANformer model (section **3.1**) – a bipartite transformer that uses multiple latent variables to modulate the evolving visual features of a generated image, so to create it in a compositional fashion.

In this paper, we explore learning from two training sets: of images and layouts. Specifically, we use panoptic segmentations [43], which specify for every segment $s_i$ its instance identity $p_i$ and semantic category $m_i$, but other types of segmentations can likewise be used. The images and layouts can either be paired or unpaired, and our training scheme accommodates both options: either by pre-training each stage independently first and then fine-tune them jointly in the paired case, or training them together from scratch in the unpaired one (See section F for further details).

### 3.1 Generative Adversarial Transformers

The GANformer [33] is a transformer-based Generative Adversarial Network: it consists of a *generator* $G(Z) = X$ that translates a partitioned latent $Z^{k \times d} = [z_1, ..., z_k]$ into an image $X^{HWc}$, and a *discriminator* $D(X)$ that seeks to discern between real and fake images. The generator begins by mapping the normally-distributed latents into an intermediate space $W = [w_1, ..., w_k]$ using a feed-forward network. Then, it sets an initial 4×4 grid, which gradually evolves through convolution and upsampling layers to the final image. Contrary to traditional GANs, the GANformer also incorporates *bipartite-transformer* layers into the generator, which compute attention between the $k$ latents and the image features, to support spatial and contentual region-wise modulation:

$$\text{Attention}(X, W) = \text{softmax}\left(\frac{q(X)k(W)^T}{\sqrt{d}}\right) v(W) \tag{1}$$

$$u(X, W) = \gamma_a(X, W) \odot LayerNorm(X) + \beta_a(X, W) \tag{2}$$

Where the equations respectively express the attention and modulation between the latents and the image; $q(\cdot), k(\cdot), v(\cdot)$ are the query, key, and value mappings; and $\gamma(\cdot), \beta(\cdot)$ compute multiplicative and additive styles (gain and bias) as a function of Attention$(X, W)$. The update rule lets the latents shape and impact the image regions as induced by the key-value attention image-to-latents assignment computed between them. This encourages visual compositionality, where different latents specialize to represent and control various semantic regions or visual elements within the generated scene.

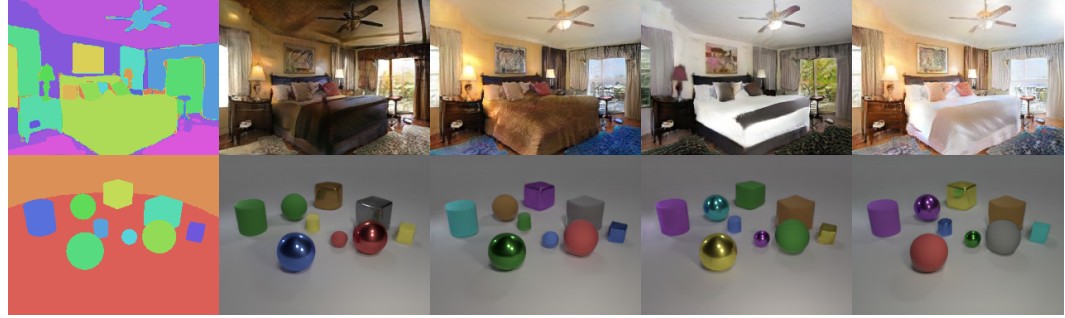

Figure 4: **GANformer2 style variation**. Images are produced by varying the style latents of the objects while keeping their structure latents fixed, achieving high visual diversity while maintaining structural consistency.

As we will see below, for each of the two generation stages, we couple a modified version of the GANformer with novel loss functions and training schemes. One key point in which we notably depart from the original GANformer regards to the number of latent variables used by the model. While the GANformer employs a fixed number of $k$ latent variables, GANformer2 supports a **variable number of latents**, thereby becoming more flexible in generating scenes of diverse complexity.

### 3.2 Planning Stage – Layout Synthesis

In the first stage, we create from scratch a schematic layout depicting the structure of the scene to be formed (see figure 2). Formally, we define a layout $S$ as a set of $k$ segments $\{s_1, ..., s_k\}$. Each generated segment $s_i$ is associated with the following[1]:

- **Shape & position**: specified by a spatial distribution $P_i(x, y)$ over the image grid $H \times W$.
- **Semantic category**: a soft distribution $M_i(c)$ over the semantic classes.
- **Unsupervised depth-ordering**: per-pixel values $d_i(x, y)$ indicating the order relative to other segments, internally used by the model to compose the segments into a layout (details below).

**Recurrent Layout Generation.** To explicitly capture conditional dependencies among segments, we construct the layout in an iterative manner, recurrently applying a parameter-lightweight GANformer $G_1$ over $T$ steps (around 2–4). At each step $t$, we generate a variable number of segments, $k_t \sim \mathcal{N}(\mu, \sigma^2)$ sampled from a trainable normal distribution[2], and gradually aggregate them into a scene layout $S_t$. To generate the segments at step $t$, we use a GANformer with $k_t$ sampled latents, each one $z_i$ is mapped through a feed-forward network $F_1(z_i) = u_i$ into an intermediate **structure latent $u_i$**, which then corresponds to segment $s_i$, guiding the synthesis of its shape, depth and position.

$$G_1(U^{k_t \times d}, S_{t-1}) = S_t \tag{3}$$

To make the GANformer recurrent, and condition the generation of new segments on ones produced in prior steps, we change the query function $q(\cdot)$ to consider not only the newly formed layout $S_t$ (as in the standard GANformer) but also the previously aggregated one $S_{t-1}$ (concatenating them together and passing as an input), allowing the model to explicitly capture dependencies across steps.

**Object Manipulation.** Notably, this construction supports a posteriori object-wise manipulation. Given a generated layout $S$, we can recreate a segment $s_i$ by modifying its corresponding latent from $z_i$ to $\hat{z}_i$ while conditioning on the other segments $S_{-i}$, mathematically by $G_1(\hat{z}_i^{1 \times d}, S_{-i})$. We thus reach a favorable balance between expressive and contextual modeling of relational interactions on the one hand, and stronger object disentanglement and controllability on the other.

**Layout Composition.** To compound the segments $s_i$ together into a layout $S_t$, we overlay them according to their predicted depths $d_i(x, y)$ and shapes $P_i(x, y)$ by computing

$$\text{Softmax}(d_i(x, y) + \log P_i(x, y))_{i=1}^k \tag{4}$$

---

[1]To produce the segments (instead of RGB values), we modify the output layer to predict for every segment its shape $P_i$, category $M_i$ and depth $d_i$. Since the distributions are soft, the model stays fully differentiable.

[2]Segments are synthesized in groups for computational efficiency when generating crowded scenes.

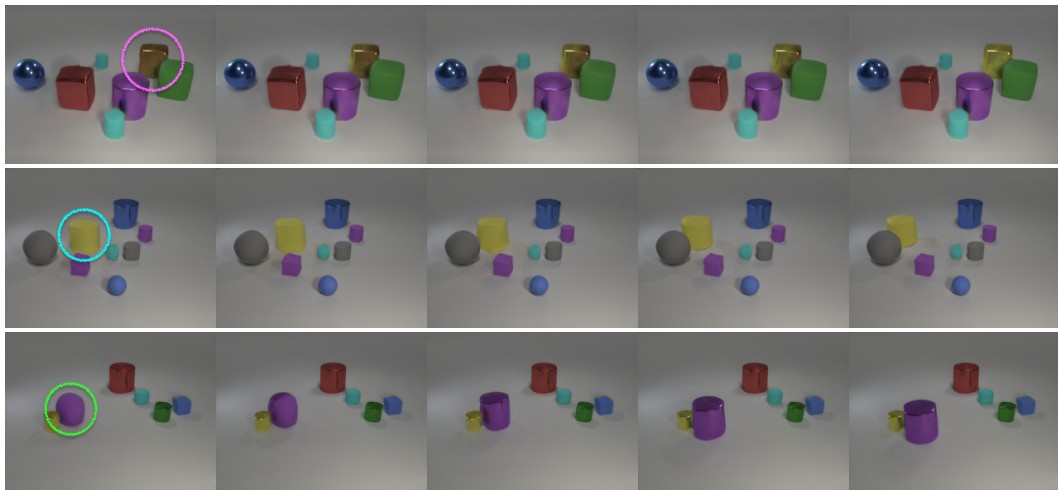

Figure 6: **Object controllability**. GANformer2 supports entity-level manipulation, and separates style from structure, enabling control over the chosen objects and properties. At each row, we begin by generating a scene from scratch (leftmost image) and then gradually interpolate the latent vector of a single object: (1) modify its **style latent** $w_i$, leading to a color change, (2) modify its **structure latent** $u_i$, resulting in a position change, and (3) modify both the structure and style latents, yielding respective changes in shape and material.

which yields for each image position $(x, y)$ a distribution over the segments. Intuitively, this allows resolving occlusions among overlapping segments. E.g. if a cube segment $i$ should appear before a sphere segment $j$, then $d_i(x, y) > d_j(x, y)$ within the overlap region $\{(x, y)\}$. Notably, we do not provide any supervision to the model about the true depths of the scene elements, and it rather creates its own internal depth ordering in an **unsupervised** manner, only indirectly moderated by the need to produce realistic looking layouts when stitching the segments together.

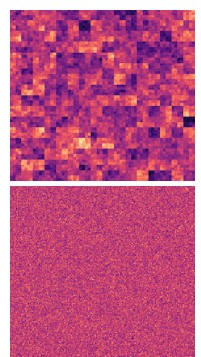

**Noise for Discrete Synthesis.** The segmentations we train the planning stage over are inherently discrete, while the generated layouts are soft and continuous. To maintain stable GAN training under these conditions, and inspired by the instance noise technique [65], we add a simple **hierarchical noise** map to the layouts fed into the discriminator, which helps ensuring its task remains challenging. We define the noise as $\sum_{r=6}^{8} \text{Upsample}(n_r^{2^r \times 2^r})$, summing up noise grids $n_r \sim \mathcal{N}(0, \sigma^2)$ across multiple resolutions, making it partly correlated between adjacent positions and more structured overall. This empirically enhances the effectiveness and substantially improves the generated layouts's quality.

Figure 5: **Hierarchical Noise scheme.**

### 3.3 Execution Stage: Layout-to-Image Translation

In the second stage, we proceed from planning to execution, and translate the generated layout $S$ into a photo-realistic image $X$. For each segment $s_i$, we first concatenate its mean depth $d_i$ and category distribution $m_i$ to its respective latent $z_i$, and map them through a feed-forward network $F_2$ into an intermediate **style latent** $w_i$, which will determine the content, texture and style of its segment. Then, we pass the resulting vectors into a second GANformer $G_2(W, S) = X$ to render the final image.

**Layout Translation.** As opposed to the original GANformer [33] that computes Attention$(X, W)$ on-the-fly to determine the assignment between latents and spatial regions, here we rely instead on the pre-planned layout $S$ as a conditional guidance source to specify this association. That way, each latent $w_i$ modulates the features within the soft region of its respective segment $s_i$, achieving a more explicit semantically-meaningful assignment between latents and scene elements. At the same time, since the whole image is still generated together, with all regions processed through the convolution, upsampling and modulation layers of $G_2$, the model can still ensure global visual coherence.

**Layout Refinement.** To increase the model's flexibility while rendering the output image, we optionally multiply the latent-to-segment distribution induced by the layout $S$ with a trainable

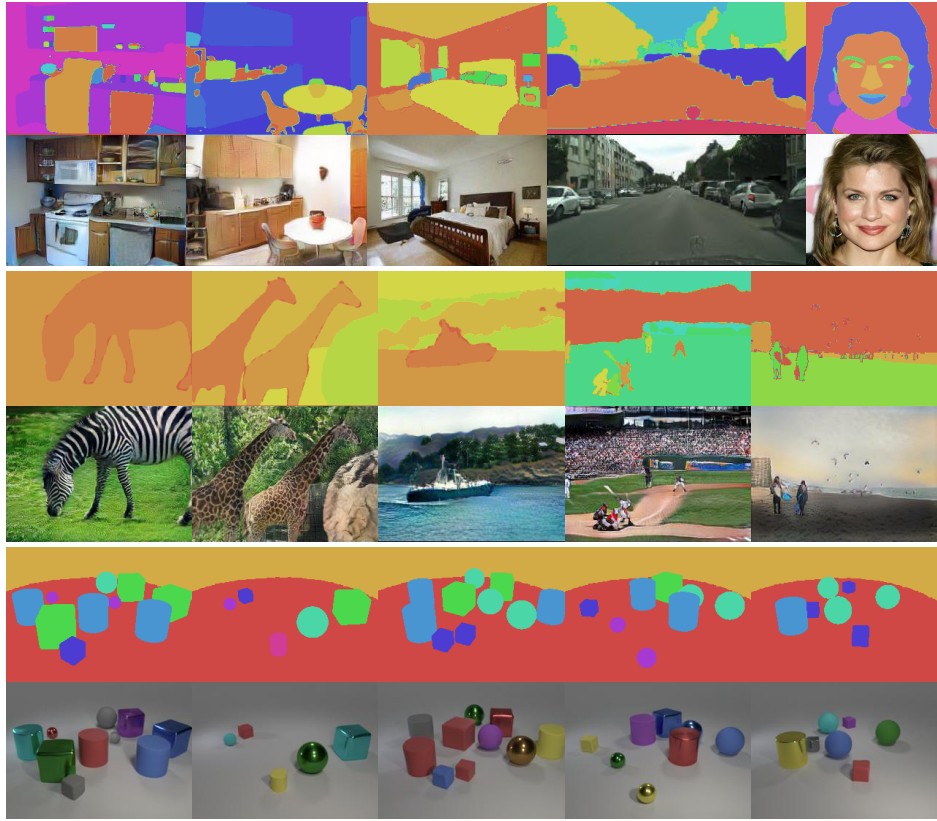

Figure 7: **Conditional generation.** Sample images generated by GANformer2 conditioned on input layouts.

*sigmoidal gate* $\sigma(g(S, X, W))$, which considers the layout $S$, the evolving image features $X$, and the style latents $W$. This allows the model to locally adjust the modulation strength as it sees fit, to refine the pre-computed layout during the final image synthesis.

**Semantic-Matching Loss.** To promote strcutrual and semantic consistency between the layout $S$ and image $X$ produced by each of the synthesis stages, we complement the standard fidelity loss $\mathcal{L}(D(\cdot))$ with two new objectives: Semantic Matching and Segment Fidelity. We note that the relation between a given layout and the set of possible images that could correspond to it is a **one-to-many relation**. To encourage it, we incorporate into the discriminator a U-Net [62] to predict the segmentation $S'$ of each image, compare it using cross-entropy to the source layout, and add the resulting loss term to the discriminator (over real samples) and to the generator (over fake ones).

$$\mathcal{L}_{SM}(S, S') \text{ where } S' = \text{Seg}(X) \tag{5}$$

This stands in a **stark contrast to the reconstruction perceptual or feature-matching losses** commonly used in conditional layout-to-image approaches [14, 35, 60], which, unjustifiably, compare the appearance of a newly generated image $X$ to that of a natural source image $X'$ (mathematically, through either $\mathcal{L}_P(X, X')$ or $\mathcal{L}_{FM}(f(X), f(X'))$), even though there is no reason for them to be similar in terms of style and texture, beyond sharing the same layout. Consequently, those losses severely and fundamentally hinder the diversity of the generated scenes (as shown in section 4). We circumvent this deficiency by comparing instead the layout implied by the new image with the source layout it is meant to follow, thereby staying true to the original objective of the translation task.

**Segment-Fidelity Loss.** To further promote the semantic alignment between the layout and the image, and encourage fidelity not only of the whole scene, but also of each segment it is composed of, we introduce the *Segment-Fidelity loss*. Given the layout $S$ and the generated image $X$, the discriminator first concatenates and processes them together through a shared convolution-downsampling stem to reduce their resolution. Then, it partitions the image according to the layout (like a jigsaw puzzle) and assesses the fidelity of each of the $n$ segments $(s_i, x_i)$ independently, through:

$$\frac{1}{n} \sum \mathcal{L}_{SF}(D(s_i, x_i)) \tag{6}$$

Table 1: **Unconditional approaches comparison.** We compare the models on generating images from scratch, using the FID and Precision/Recall metrics for evaluation, computed over 50k sample images. $COCO_p$ refers to a mean score over a partitioned version of the dataset, created to improve samples' visual quality (section I).

| Model | CLEVR | | | Bedrooms | | | FFHQ | | |
|---|---|---|---|---|---|---|---|---|---|
| | FID ↓ | Precision | Recall | FID ↓ | Precision | Recall | FID ↓ | Precision | Recall |
| GAN | 25.02 | 21.77 | 16.76 | 12.16 | 52.17 | 13.63 | 13.18 | 67.15 | 17.64 |
| k-GAN | 28.29 | 22.93 | 18.43 | 69.90 | 28.71 | 3.45 | 61.14 | 50.51 | 0.49 |
| SAGAN | 26.04 | 30.09 | 15.16 | 14.06 | 54.82 | 7.26 | 16.21 | 64.84 | 12.26 |
| StyleGAN2 | 16.05 | 28.41 | 23.22 | 11.53 | 51.69 | 19.42 | 10.83 | 68.61 | 25.45 |
| VQGAN | 32.60 | 46.55 | 63.33 | 59.63 | 55.24 | 28.00 | 63.12 | 67.01 | 29.67 |
| SBGAN | 35.13 | 48.12 | 64.41 | 48.75 | 56.14 | 31.02 | 24.52 | 58.32 | 8.17 |
| GANformer | 9.17 | 47.55 | 66.63 | 6.51 | 57.41 | 29.71 | **7.42** | **68.77** | 5.76 |
| **GANformer2** | **4.70** | **64.18** | **67.03** | **6.05** | **60.93** | **37.15** | 7.77 | 61.54 | **34.45** |

| Model | COCO | | | $COCO_p$ | | | Cityscapes | | |
|---|---|---|---|---|---|---|---|---|---|
| | FID ↓ | Precision | Recall | FID ↓ | Precision | Recall | FID ↓ | Precision | Recall |
| GAN | 41.00 | 48.57 | 7.06 | 37.53 | 51.02 | 9.30 | 11.56 | **61.09** | 15.30 |
| k-GAN | 63.51 | 42.06 | 5.42 | 61.05 | 45.37 | 7.04 | 51.08 | 18.80 | 1.73 |
| SAGAN | 46.09 | 50.00 | 7.47 | 42.05 | 51.71 | 7.84 | 12.81 | 43.48 | 7.97 |
| StyleGAN2 | 26.79 | 50.92 | 23.30 | 25.24 | 52.93 | 24.48 | 8.35 | 59.35 | 27.82 |
| VQGAN | 63.12 | 53.05 | 27.22 | 65.20 | **55.42** | 30.12 | 173.8 | 30.74 | 43.00 |
| SBGAN | 108.1 | 42.53 | 17.53 | 104.32 | 44.43 | 19.12 | 54.92 | 52.12 | 25.08 |
| GANformer | 25.24 | **53.68** | 12.31 | 23.81 | 55.29 | 14.48 | **5.76** | 48.06 | 33.65 |
| **GANformer2** | **21.58** | 49.12 | **29.03** | **20.41** | 51.07 | **32.84** | 6.21 | 56.12 | **54.18** |

Using the stem both reduces the dimensiononality to improve computational efficiency, and further allows for contextual information of each segment's surroundings to propagate into its representation. This improves over the stricter patchGAN [35] which blindly splits the image into a fixed tabular grid, while our new loss offers a more natural choice, dividing it up into semantically-meaningful regions.

## 4 Experiments

We investigate GANformer2's properties through a suite of quantitative and qualitative experiments. As shown in **section 4.1**, the model attains state-of-the-art performance, successfully generating images of high quality, diversity and semantic consistency, over multiple challenging datasets of highly-structured simulated and real-world scenes. In **section 4.2**, we compare the execution stage to leading layout-to-image approaches, demonstrating significantly larger output variation, possibly thanks to the new structural loss functions we incorporate into the network. Further analysis conducted in **sections 4.3 and 4.4** reveals several favorable properties that GANformer2 possesses, specifically of enhanced interpretabilty and strong spatial disentanglement, which enables object-level manipulation. In the **supplementary**, we provide additional visualizations of style and structure disentanglement as well as ablation and variation studies, to shed more light upon the model behavior and operation. Taken altogether, the evaluation offers solid evidence for the effectiveness of our approach in modeling compositional scenes in a robust, controllable and interpretable manner.

### 4.1 State-of-the-Art Comparison

We compare our model with both baselines as well as leading approaches for visual synthesis. Unconditional models include a baseline GAN [23], StyleGAN2 [42], Self-Attention GAN [81], the autoregressive VQGAN [21], the layerwise k-GAN [69], the two-stage SBGAN [5] and the original GANformer. Conditional approaches include SBGAN, BicycleGAN [86], Pix2PixHD [72], and SPADE [60]. We implement the unconditional methods within our public GANformer codebase and use the authors' official implementations for the conditional ones. All models have been trained with resolution of $256 \times 256$ and for an equal number of training steps, roughly spanning 10 days on a single V100 GPU per model. See section H for description of baselines and competing approaches, implementation details, hyperparameter settings, data preparations and training configuration.

As tables 1 and 3 show, the GANformer2 model outperforms prior work along most metrics and datasets in both the conditional and unconditional settings. It achieves substantial gains in terms of FID score, which correlates with visual quality [29], as is likewise reflected through the good Precision scores. Notable also are the large improvements along Recall, in the unconditional and especially the conditional settings, which indicate wider coverage of the natural image distribution. We note that the performance gains are highest for the CLEVR [37] and the COCO [48] datasets, both focusing on varied arrangements of multiple objects and high structural variance. These results serve as a strong evidence for the particular aptitude of GANformer2 for compositional scenes.

Table 2: **Disentanglement & controllability comparison** over sample CLEVR images, using the DCI metrics for latent-space disentanglement and our correlation-based metrics for controllability (see section 4.4).

| Model | Disentanglement | Modularity | Completeness | Informativeness | Informativeness' | Object-$\rho \downarrow$ | Property-$\rho \downarrow$ |
|---|---|---|---|---|---|---|---|
| GAN | 0.126 | 0.631 | 0.071 | 0.583 | 0.434 | 0.72 | 0.85 |
| StyleGAN | 0.208 | 0.703 | 0.124 | 0.685 | 0.332 | 0.64 | 0.49 |
| GANformer | 0.768 | 0.952 | 0.270 | 0.972 | 0.963 | 0.38 | 0.45 |
| MONet | 0.821 | 0.912 | 0.349 | 0.955 | 0.946 | 0.29 | 0.37 |
| Iodine | 0.784 | **0.948** | 0.382 | 0.941 | 0.937 | 0.27 | 0.35 |
| **GANformer2** | **0.852** | 0.946 | **0.413** | **0.974** | **0.965** | **0.19** | **0.32** |

## 4.2 Scene Diversity & Semantic Consistency

Beyond visual fidelity, we study the models under the potentially competing aims of content variability on the one hand and semantic consistency on the other. To assess consistency, we compare the input layouts $S$ with those implied by the synthesized images, using standard metrics of mean IoU, pixel accuracy and ARI. Following prior works [60, 86], the inferred layouts $S'$ are obtained by a pre-trained segmentation model that we use for evaluation purposes. To measure diversity, we sample $n = 20$ images $X_i$ for each input layout $S$, and compute the mean LPIPS pairwise distance [82] over image pairs $X_i, X_j$. Ideally, we expect a good model to achieve strong alignment between each input layout and the one derived from the output synthesized image, while still maintaining high variance between samples that are conditioned on the same source layout.

We see in table 3 that GANformer2 produces images of significantly better diversity, as reflected both through the LPIPS and Recall scores. Other conditional approaches such as Pix2PixHD, BicycleGAN and SPADE that rely on pixel-wise perceptual or feature matching, reach high precision at the expense of considerably reduced variability, revealing the weakness of those losses. We further note that our model achieves better consistency than prior work, as is also illustrated by figures 22-24 which present qualitative comparison between synthesized samples. These results demonstrate the benefits of our new structural losses (section 3.3), as is also corroborated by the model's learning efficiency when evaluated with different objectives (figure 8). We compare the Semantic-Matching and Segment-Fidelity losses with several alternatives (section D), and observe improved performance and accelerated learning when using the new objectives, possibly thanks to the local semantic alignment and finer-detail fidelity they respectively encourage.

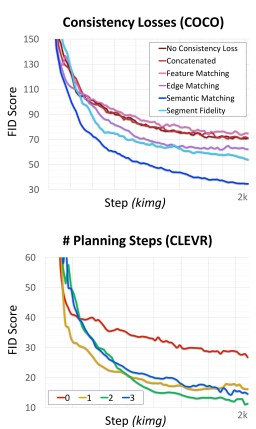

Figure 8: **Model's learning curves.**

## 4.3 Transparency & Interpretability

An advantage of our model compared to traditional GANs, is that we incorporate explicit structure into the visual generative process, which both makes it more explainable, while providing means for controlling individual scene elements (section 4.4). In figures 3 and 11, we see how GANformer2 gradually constructs the image layout, recurrently adding new object segments into the scene. The gains of iterative over non-iterative generation are also demonstrated by figures 8 and 21. We note that different datasets benefit from a different number of recurrent synthesis steps, reflecting their respective degree of structural compositionality (section E). Figures 2 and 10 further show that the model develops an internal notion for segments' depth, which aids it in determining the most effective order to compile them together. Remarkably, no explicit information is given to the network about the depth of entities within the scene, nor about effective orders to create them, and it rather learns these on its own in an unsupervised manner, through the indirect need to identify relative depths and segment ordering that will enable creating feasible and compelling layouts.

## 4.4 Controllability & Disentanglement

The iterative nature of GANformer2 and its compositional latent space enhance its disentanglement along two key dimensions: **spatially**, enabling independent control over chosen objects, and **semantically**, separating between structure and style – the former is considered during planning while the latter at execution. Figures 6 and 13-16 feature examples where we modify an object of interest, changing its color, material, shape, or position, all without negatively impacting its surroundings. In figures 4 and 17-20, we produce images of diverse colors and textures while conforming to a source layout, and conversely, vary the scene composition while maintaining a consistent style. Finally,

Table 3: **Conditional approaches comparison.** We compare conditional generative models that map input layouts into output images. Multiple metrics are used for different purposes: FID, Inception (IS) and Precision scores for visual quality; class-weighted mean IoU, Pixel Accuracy (pAcc) and Adjusted Rand Index (ARI) for layout-image consistency; Recall for coverage of the natural image distribution; and LPIPS for measuring the output visual diversity given an input layout. All metrics are computed over 50k samples per model.

| CLEVR | | | | | | | |
|---|---|---|---|---|---|---|---|
| **Model** | FID ↓ | IS ↑ | Precision ↑ | Recall ↑ | mIoU ↑ | pAcc ↑ | ARI ↑ | LPIPS ↑ |
| Pix2PixHD | 4.88 | 2.19 | 72.49 | 60.40 | 98.18 | 99.06 | 97.90 | 6e-8 |
| SPADE | 5.74 | 2.23 | 70.25 | 61.96 | **98.30** | 99.12 | 98.01 | 0.12 |
| BicycleGAN | 35.62 | 2.29 | 6.27 | 10.11 | 88.15 | 92.78 | 90.21 | 0.10 |
| SBGAN | 18.02 | 2.22 | 52.29 | 48.20 | 95.27 | 97.81 | 92.34 | 8e-7 |
| **GANformer2** | **0.99** | **2.34** | **73.26** | **78.38** | 97.83 | **99.15** | **98.46** | **0.19** |

| Bedrooms | | | | | | | |
|---|---|---|---|---|---|---|---|
| **Model** | FID ↓ | IS ↑ | Precision ↑ | Recall ↑ | mIoU ↑ | pAcc ↑ | ARI ↑ | LPIPS ↑ |
| Pix2PixHD | 32.44 | 2.27 | 70.46 | 2.41 | 64.54 | 76.62 | 75.48 | 2e-8 |
| SPADE | 17.06 | 2.33 | **80.20** | 7.04 | 70.27 | 80.63 | 79.44 | 0.07 |
| BicycleGAN | 23.34 | 2.52 | 49.68 | 10.45 | 58.47 | 70.76 | 72.73 | 0.22 |
| SBGAN | 29.35 | 2.28 | 56.21 | 7.93 | 65.31 | 71.39 | 73.24 | 3e-8 |
| **GANformer2** | **5.84** | **2.55** | 54.92 | **41.94** | **79.06** | **86.62** | **83.00** | **0.50** |

| CelebA | | | | | | | |
|---|---|---|---|---|---|---|---|
| **Model** | FID ↓ | IS ↑ | Precision ↑ | Recall ↑ | mIoU ↑ | pAcc ↑ | ARI ↑ | LPIPS ↑ |
| Pix2PixHD | 54.47 | 2.44 | 57.48 | 0.73 | 68.04 | 80.18 | 63.75 | 4e-9 |
| SPADE | 33.42 | 2.35 | **85.08** | 2.72 | 67.99 | 80.11 | 63.80 | 0.07 |
| BicycleGAN | 56.56 | 2.53 | 52.93 | 2.36 | 66.95 | 79.41 | 62.50 | 0.15 |
| SBGAN | 50.92 | 2.44 | 59.28 | 3.62 | 67.42 | 80.12 | 61.42 | 4e-9 |
| **GANformer2** | **6.87** | **3.17** | 57.32 | **37.06** | **68.88** | **80.66** | **64.65** | **0.38** |

| Cityscapes | | | | | | | |
|---|---|---|---|---|---|---|---|
| **Model** | FID ↓ | IS ↑ | Precision ↑ | Recall ↑ | mIoU ↑ | pAcc ↑ | ARI ↑ | LPIPS ↑ |
| Pix2PixHD | 47.74 | 1.51 | 50.53 | 0.34 | 73.20 | 83.49 | 78.48 | 3e-8 |
| SPADE | 22.98 | 1.46 | 83.32 | 9.55 | 75.96 | 85.25 | **80.74** | 3e-3 |
| BicycleGAN | 25.02 | 1.56 | 63.01 | 9.49 | 67.24 | 78.43 | 72.20 | 7e-4 |
| SBGAN | 45.82 | 1.52 | 51.38 | 5.39 | 72.04 | 81.90 | 79.32 | 4e-8 |
| **GANformer2** | **5.95** | **1.63** | **84.12** | 37.72 | **76.11** | **85.58** | 80.12 | **0.27** |

| COCO | | | | | | | |
|---|---|---|---|---|---|---|---|
| **Model** | FID ↓ | IS ↑ | Precision ↑ | Recall ↑ | mIoU ↑ | pAcc ↑ | ARI ↑ | LPIPS ↑ |
| Pix2PixHD | 18.91 | 17.97 | **80.91** | 16.35 | 58.75 | 71.94 | 68.20 | 1e-8 |
| SPADE | 22.80 | 16.29 | 79.00 | 10.63 | 55.72 | 69.55 | 67.73 | 0.19 |
| BicycleGAN | 69.88 | 9.16 | 40.05 | 5.83 | 33.58 | 47.98 | 56.55 | 0.18 |
| SBGAN | 45.94 | 14.31 | 58.93 | 10.27 | 47.82 | 54.58 | 62.29 | 4e-8 |
| **GANformer2** | **17.39** | **18.01** | 70.20 | **24.86** | **74.93** | **84.37** | **76.41** | **0.38** |

as illustrated by figures 9 and 12, we can even add or remove objects to the synthesized scene, while respecting interactions of occlusions, shadows and reflections, effectively achieving amodal completion, and potentially extrapolating beyond the training data (see sections C-B).

To quantify the disentanglement degree, we refer to the DCI and modularity metrics [17, 83], which measure the correspondence between latent factors and image attributes. As in [33, 75], we consider a sample set of synthesized CLEVR images, using a pre-trained object detector to extract the objects and attributes. To further assess the controllability, we perform random perturbations over the latent variables used to generate the scenes, and estimate the mean correlation between the resulting semantic changes among object and property pairs. Intuitively, we expect a disentangled representation to yield uncorrelated changes among objects and attributes. As shown in table 2, GANformer2 attains higher disentanglement and stronger controllability than leading GAN-based and variational approaches, quantitatively confirming the efficacy of our approach.

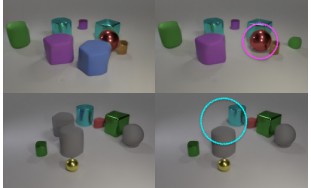

Figure 9: **Amodal completion** for object removal.

## 5   Conclusion

In this paper, we have tackled the task of visual synthesis, and introduced the GANformer2 model, an iterative transformer for transparent and controllable scene generation. Similar to how a person may approach the creative task, it first sketches a high-level outline and only then fills in the details. By equipping the network with a strong explicit structure, we endow it with the capacity to plan ahead its generation process, capture the dependencies and interactions between objects within the scene, and reason about them directly, moving a step closer towards compositional generative modeling.

# 6 Acknowledgements

We performed the experiments for the paper on AWS cloud, thanks to Stanford HAI credit award. Drew A. Hudson (Dor) is a PhD student at Stanford University and C. Lawrence Zitnick is a research scientist at Facebook FAIR. We wish to thank the anonymous reviewers for their thorough, insightful and constructive feedback, questions and comments.

*Dor wishes to thank Prof. Christopher D. Manning for the kind PhD support over the years that allowed this work to happen!*

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
