# Compositional Transformers for Scene Generation
## *Supplementary Material*

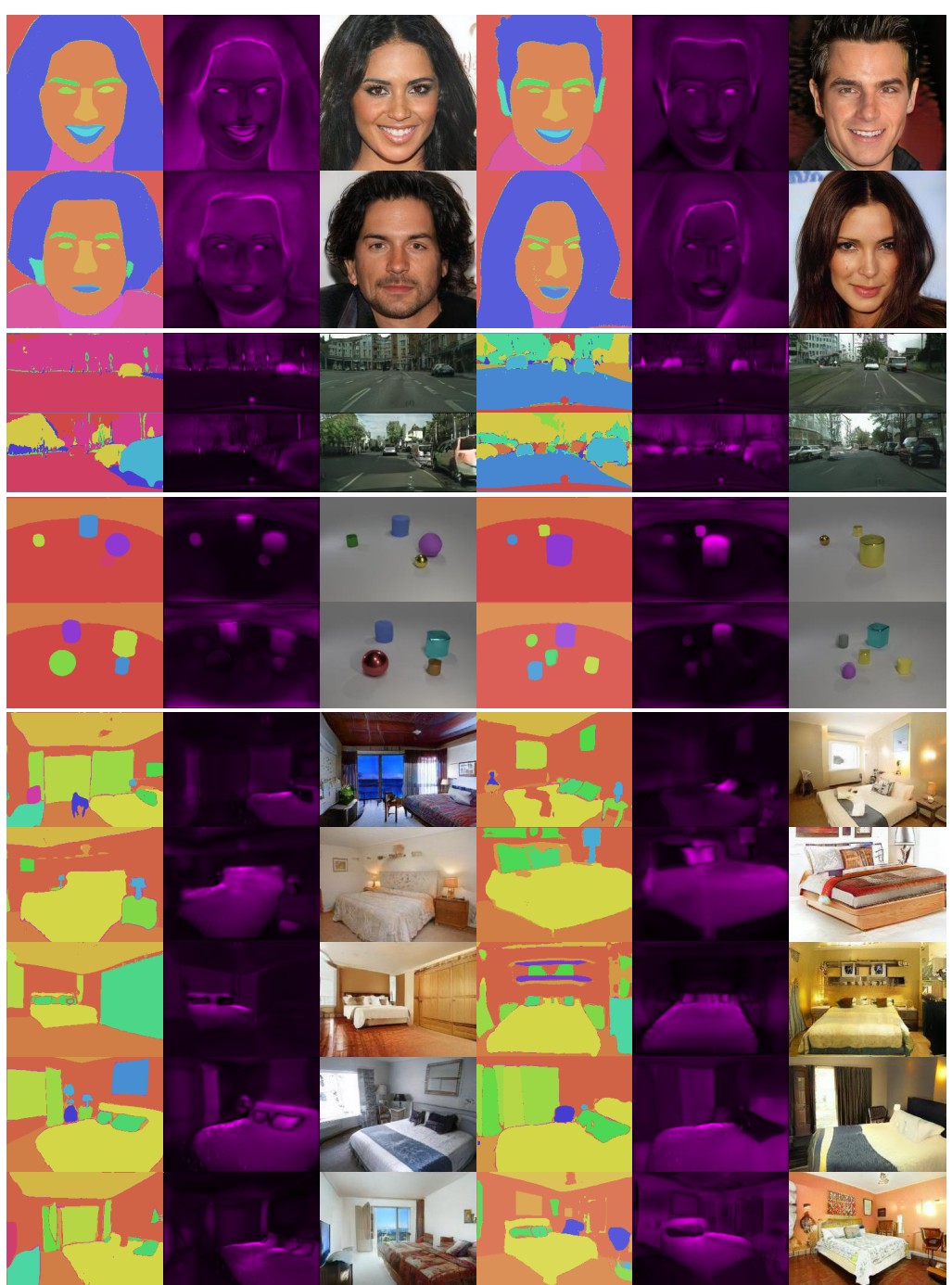

Figure 10: A visualization of the **layouts and unsupervised depth maps** produced by GANformer2's planning stage while synthesizing varied images, making the generative process more structured and interpretable.

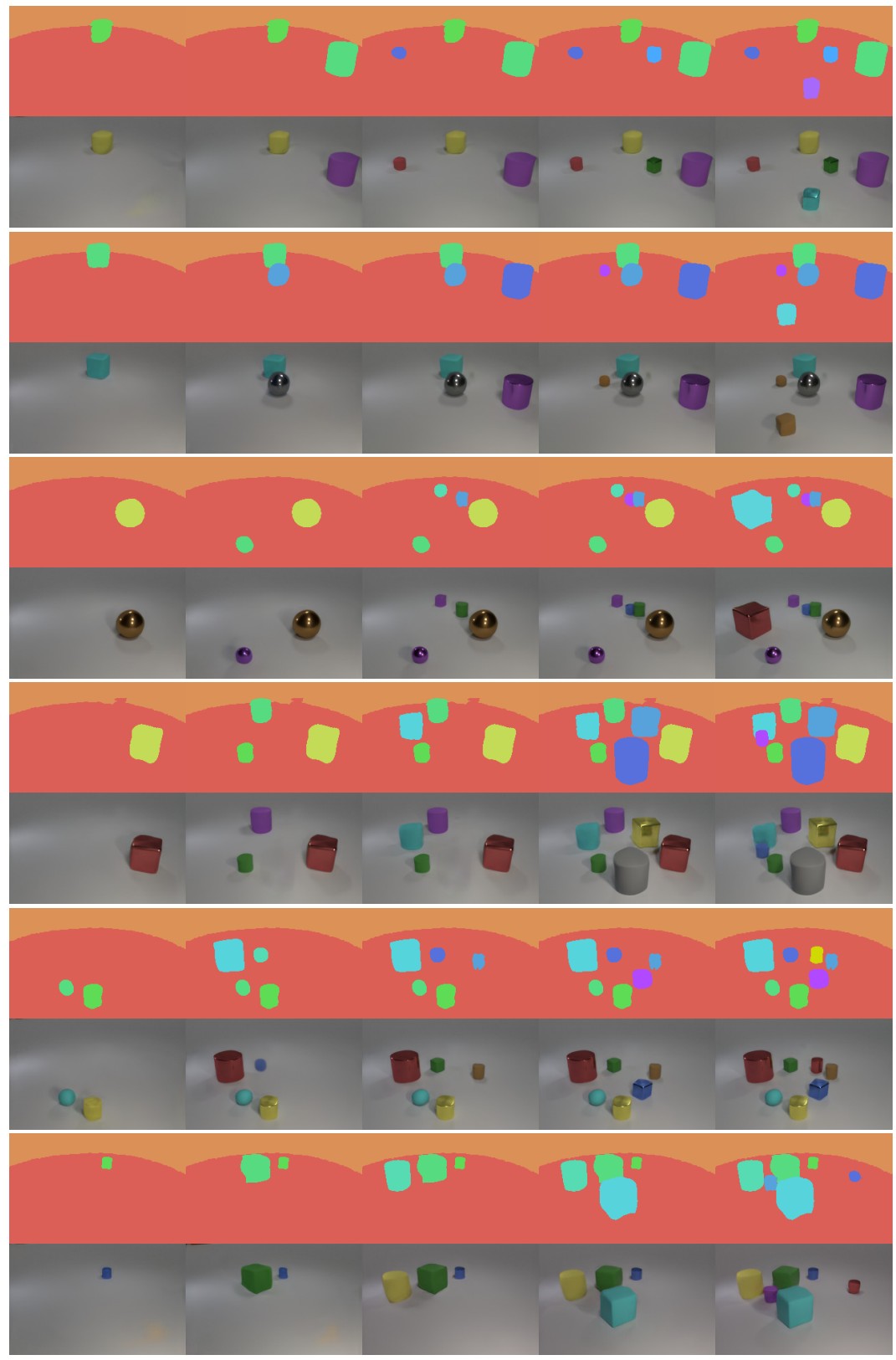

Figure 11: **Recurrent scene generation.** GANformer2 creates the layout sequentially, segment-by-segment, to capture the scene's compositionality, effectively allowing us to add or remove objects from the resulting images.

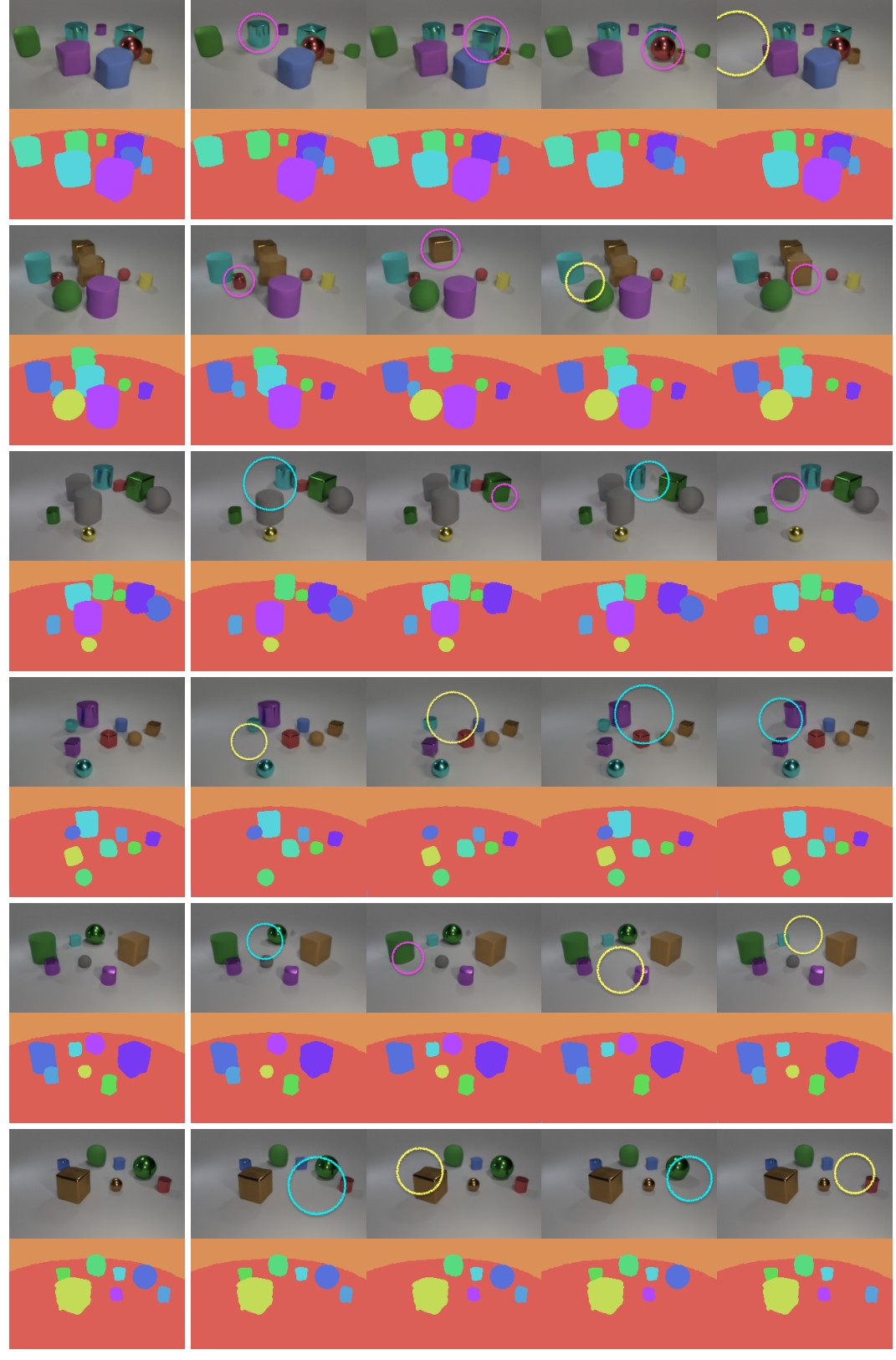

Figure 12: **Object removal**. Since GANformer2 creates each scene as a composition of interacting segments, it supports adding and removal of objects while respecting various dependencies with their surroundings: Amodal completion of occluded objects is denoted by **pink**, updates of shadows and especially reflections by **cyan**, and other object removals cases by **yellow**.

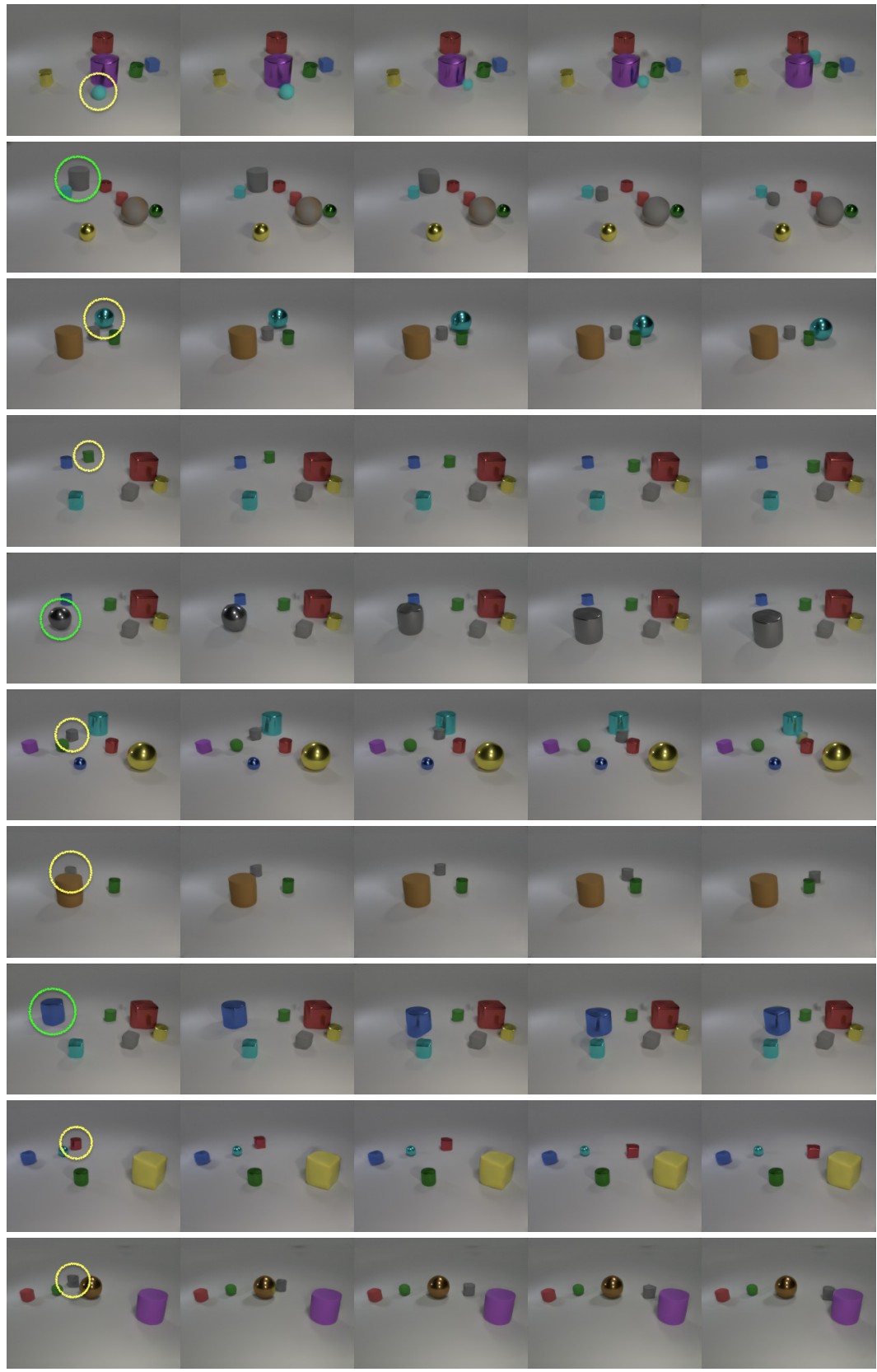

Figure 13: **Object controllability over structural attributes**, achieved by modifying the model's structure latent $u_i$ of the chosen object during the planning stage (section 3.2). Shape manipulation is denoted by **green**, while position changes by **yellow**.

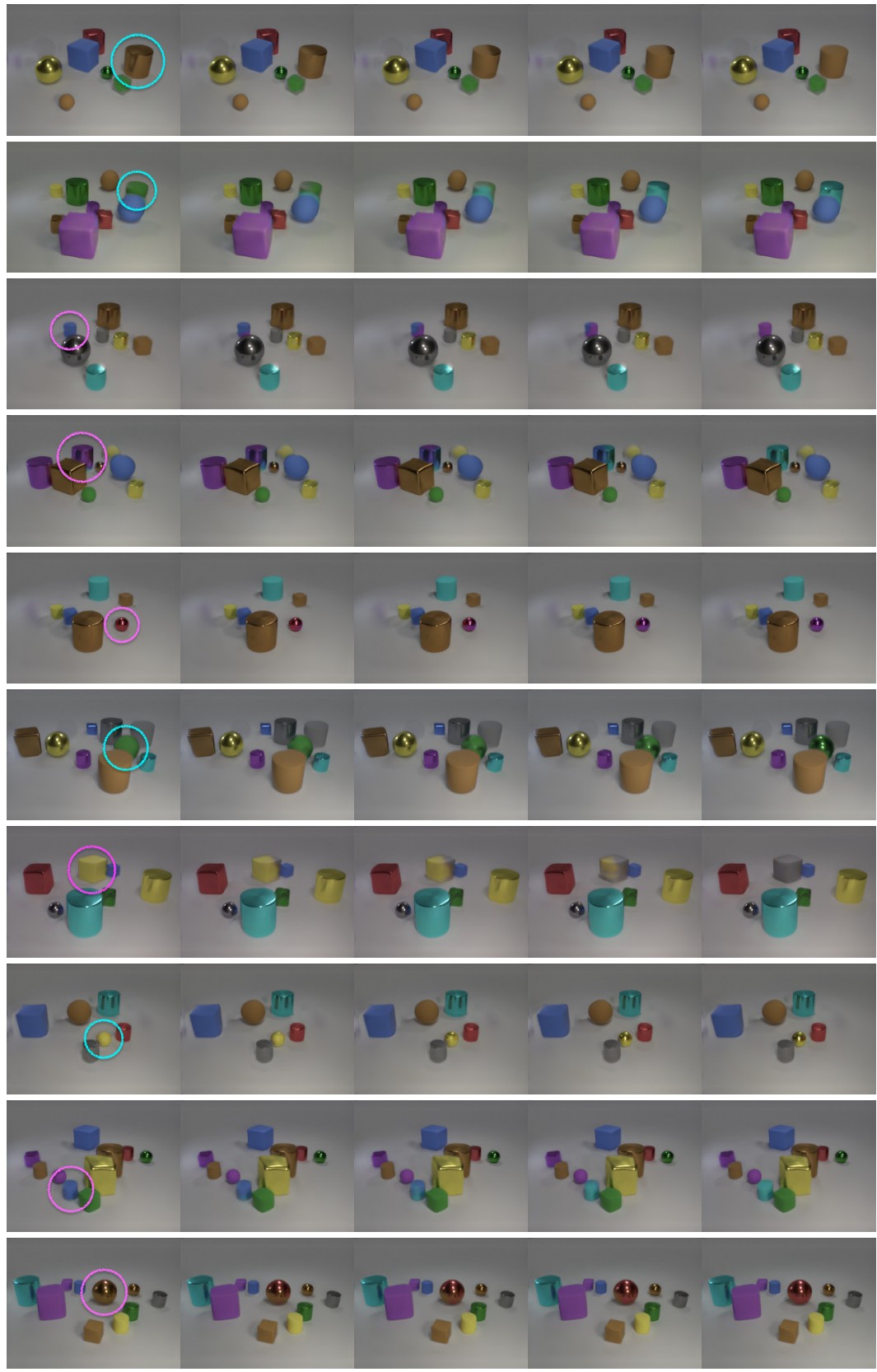

Figure 14: **Object controllability over stylistic attributes**, achieved by modifying the model's style latent $w_i$ of the chosen object during the execution stage (section 3.3). Color manipulation is denoted by **pink**, while updates of material by **cyan**.

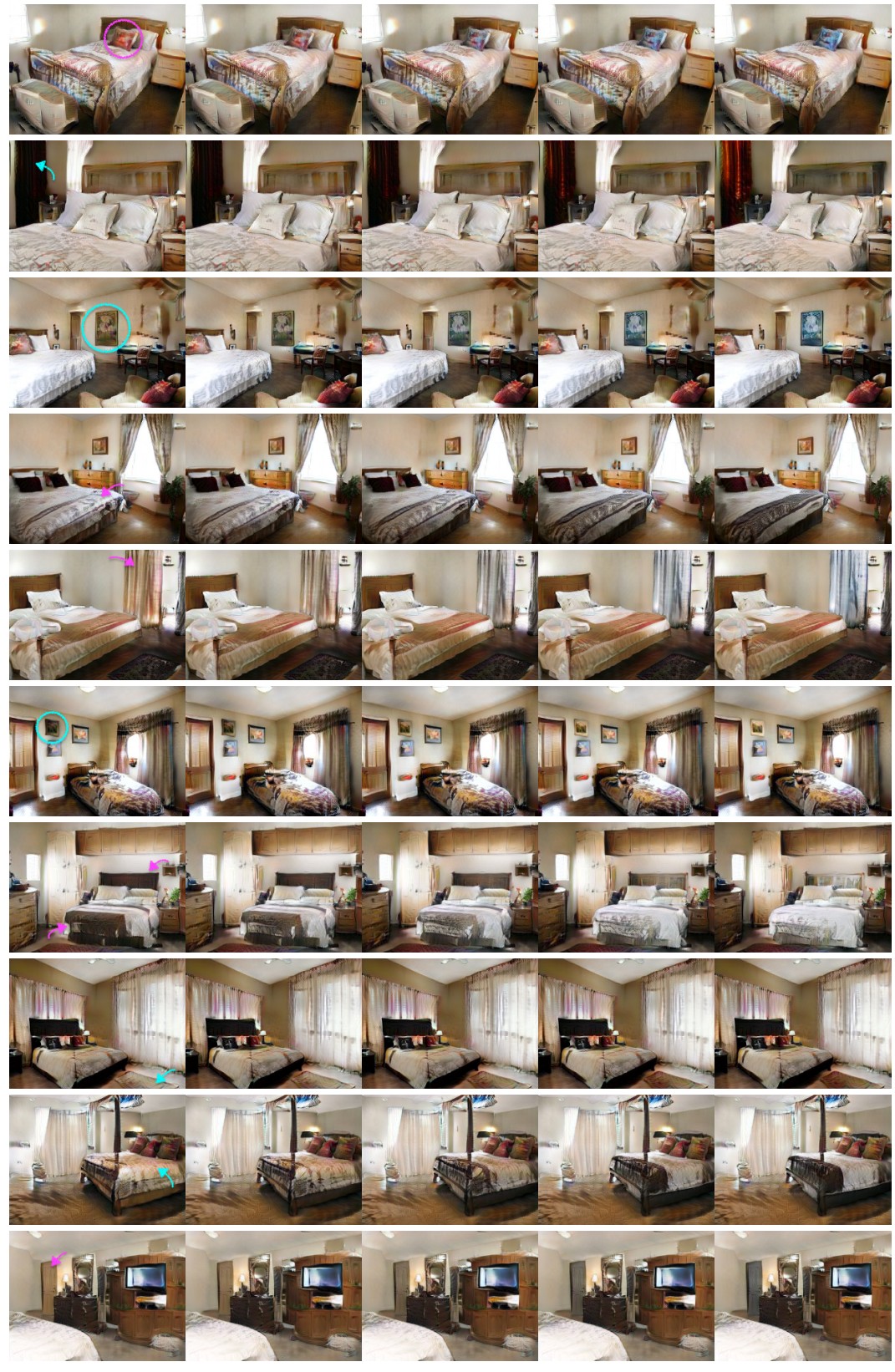

Figure 15: **Localized property manipulation** of selected objects without negatively impacting their surroundings, over LSUN-Bedrooms scenes.

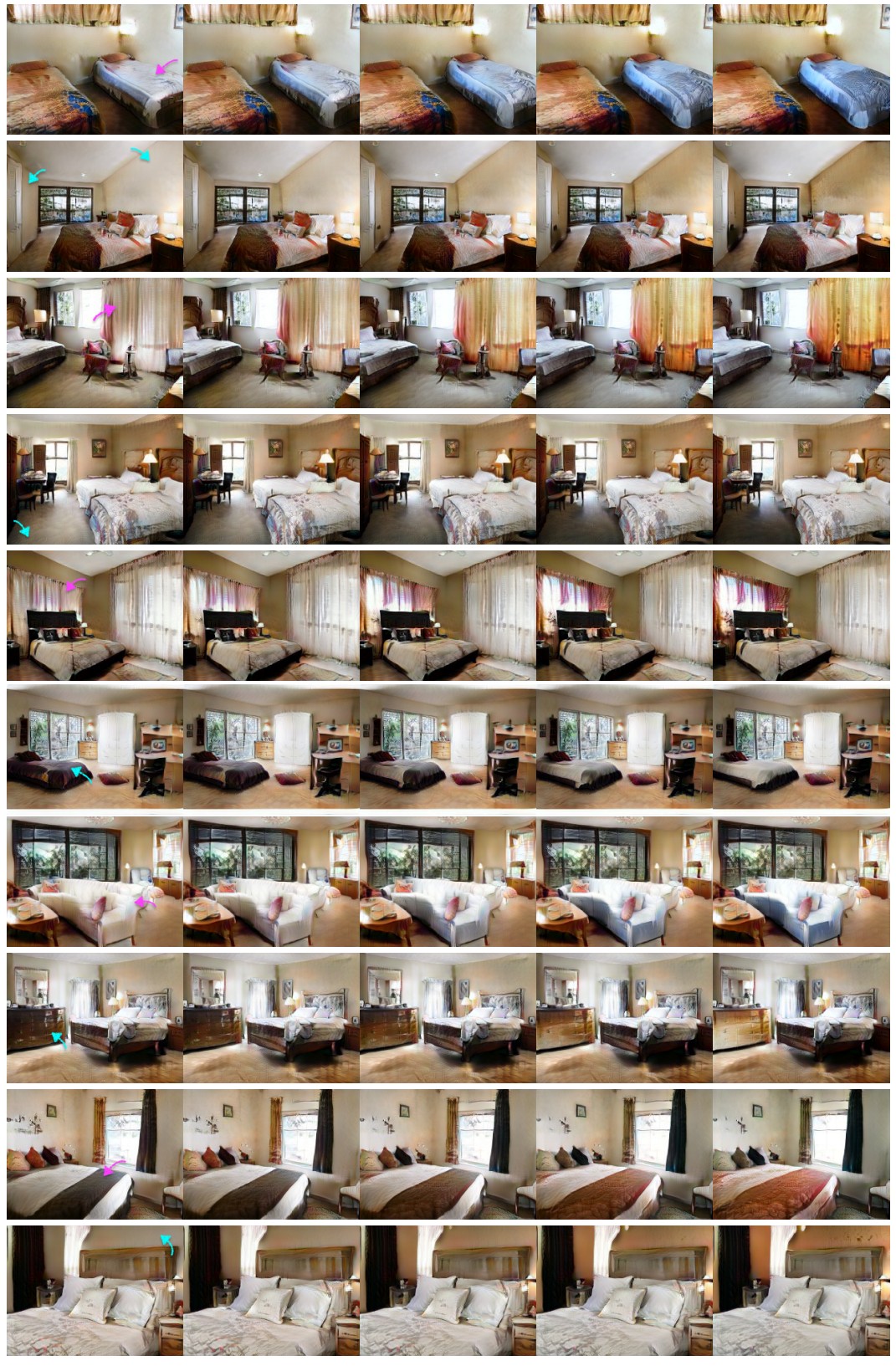

Figure 16: **Localized property manipulation** of selected objects, without negatively impacting their surroundings over LSUN-Bedrooms scenes (***continued***).

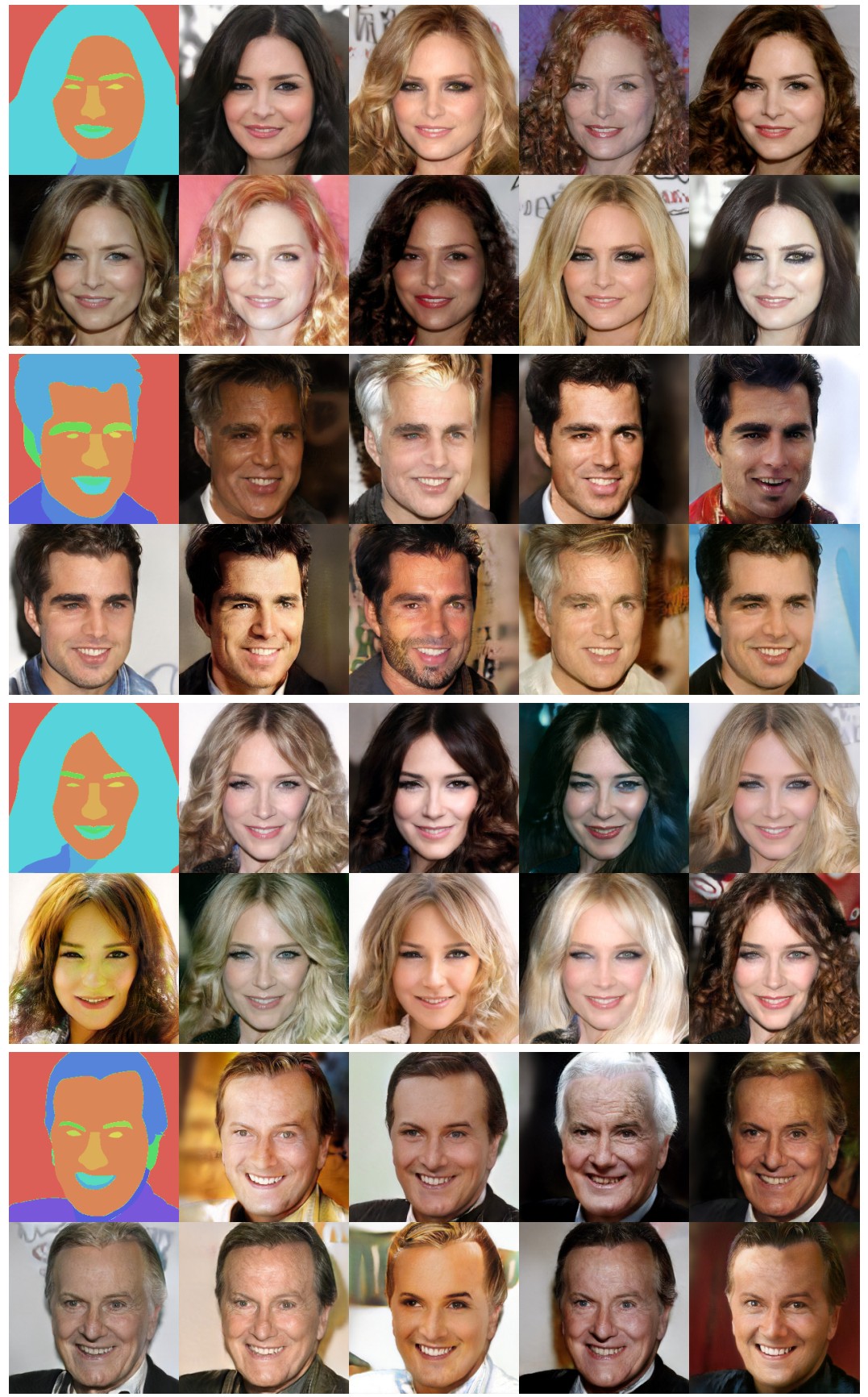

Figure 17: **GANformer2 conditional generative diversity** for CelebA. Images conform to the source layouts while still demonstrating high variance, featuring diversity both in stylistic aspects of lighting conditions and color scheme, but also in structural ones, as reflected through the hair type, background and age.

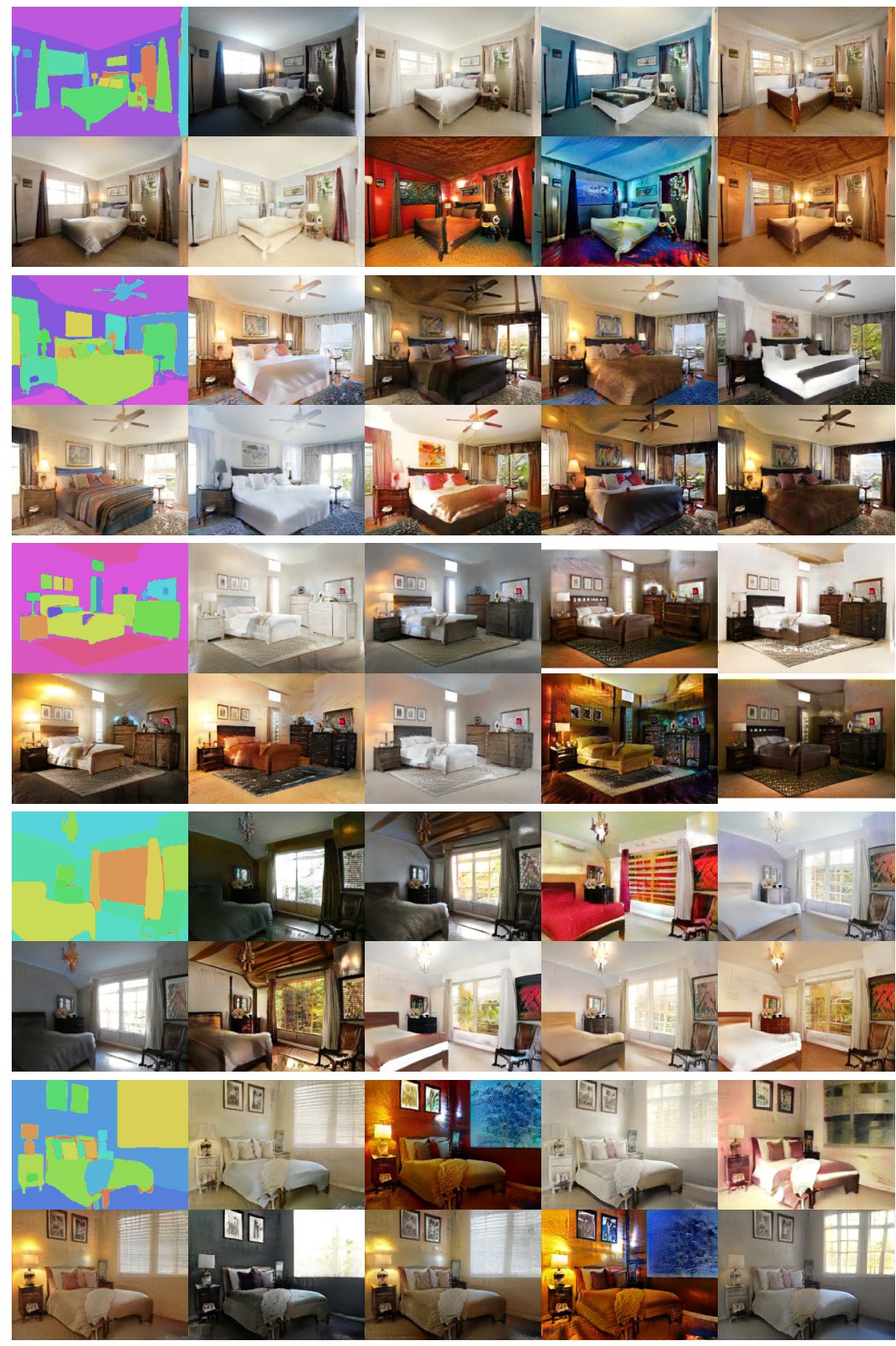

Figure 18: **GANformer2 conditional generative diversity** for LSUN-Bedrooms. Images conform to the source layouts on the one end while still demonstrating high variance on the other, featuring diversity both in stylistic aspects of lighting condition and color scheme, but also in structural ones, as reflected through the windows, paintings and bedding.

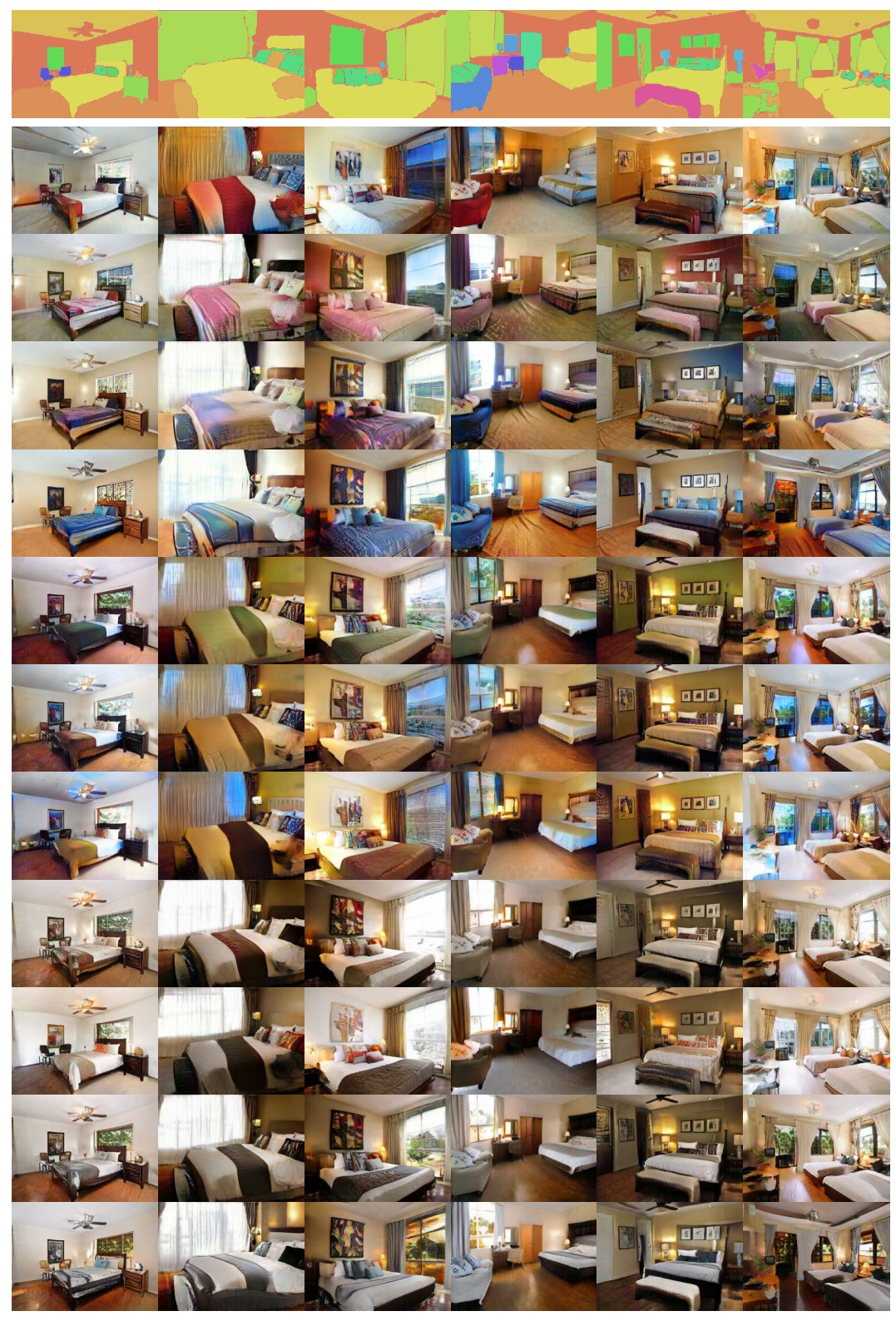

Figure 19: **GANformer2 style and structure separation**. We manipulate each aspect while maintaining consistency over the other, varying the structure between columns and style between rows.

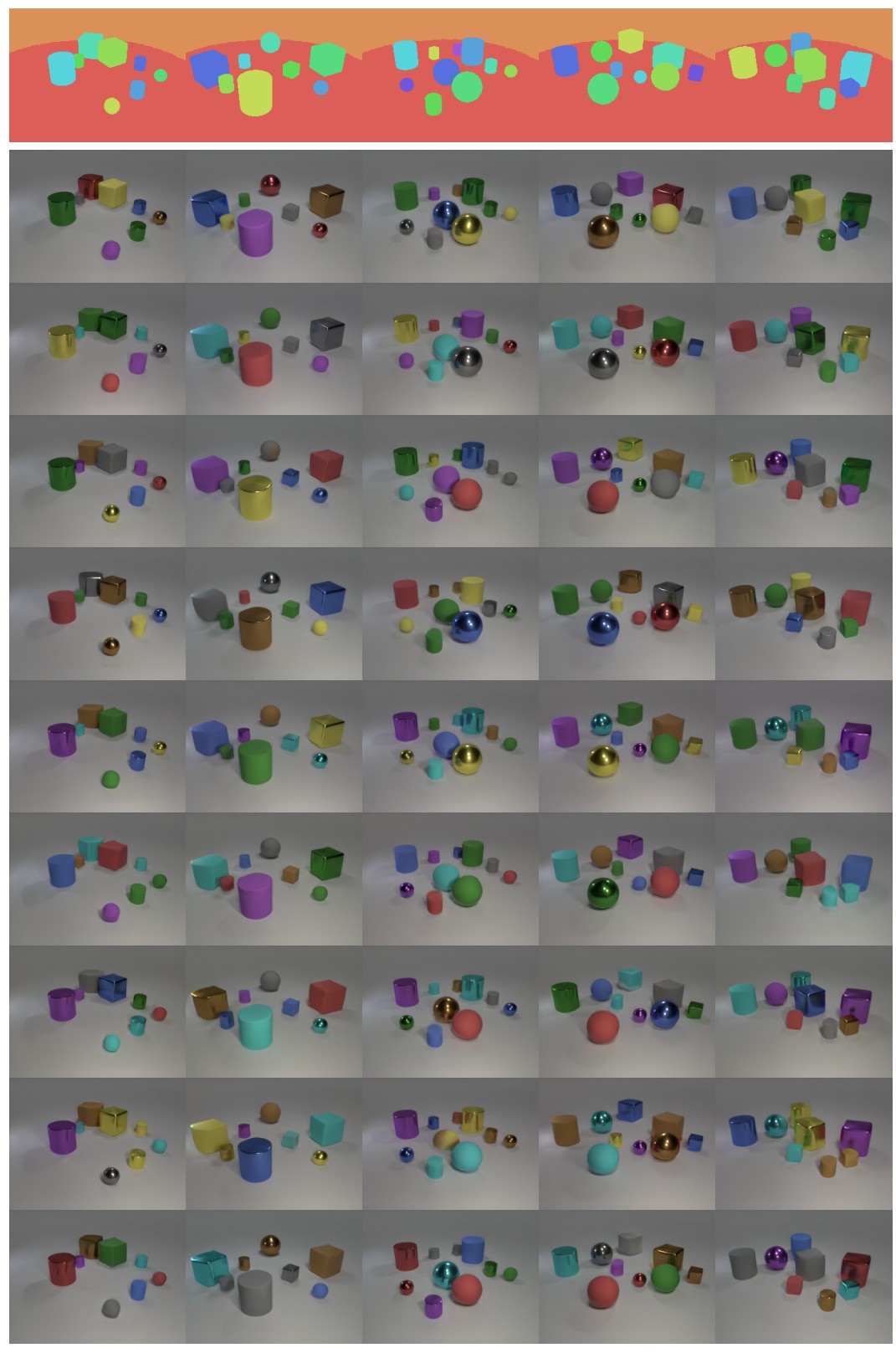

Figure 20: **GANformer2 conditional generative diversity** for CLEVR. The scenes significantly vary in combinations of objects' colors and materials (contrary to competing approaches, as shown in figures 23-24, while still closely following the source layouts.

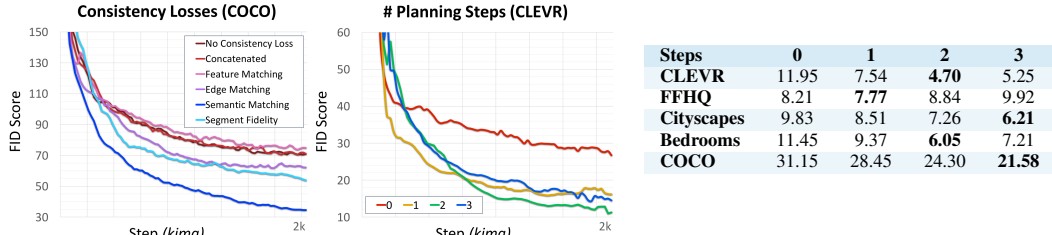

Figure 21: **Learning curves and performance comparison** when varying the loss function for the execution stage **(left)**, and the number of recurrent planning steps for the layout synthesis **(center, right)**. In each step, a random number of segments is generated, sampled from a trainable normal distribution, to flexibly model highly-structured scenes and capture conditional object dependencies within them while still maintaining good computational efficiency. "0" denotes a non-compositional model that generates the layouts in a single pass rather than as a collection of segments.

## A Overview

In the following, we provide additional qualitative and quantitative experiments for the GANformer2 model. **Figure 10** shows visualizations of the model's generative process as it produces depth-aware scene layouts which in turn guide and facilitate the synthesis of output photo-realistic images. The model's compositional structure does not only enhance its transparency, but also increases its controllability (**section B**, **figures 13-16**), allowing GANformer2 to manipulate properties of individual objects without negatively altering their surroundings, and even develop capacity for amodal completion of occluded objects (**section C**, **figures 11-12**). The obtained decoupling between structure and style is further illustrated through **figures 17-18** and **19-20**, respectively demonstrating variability along one aspect while maintaining consistency over the other.

We believe the diversity achieved by GANformer2 results from the new structural losses we introduce (**section 3.3**), which we compare to several baselines and alternatives in **figures 22-24** and **section D**. In **section E**, we proceed through additional model ablation and variation studies, to asses the contribution of its different architectural components, and the number of recurrent planning steps in particular. **Section F** focuses on the training configuration, comparing and contrasting between paired vs. unpaired settings. We conclude the supplementary by providing implementation details (**section G**), description of baselines and competing methods (**section H**), and information about data preparation procedures (**section I**).

## B Style & Structure Disentanglement

GANformer2 decomposes the synthesis into two stages of planning and execution, the former produces the scene layout and structure while the latter controls its texture, colors and style. Figures figures 17-18 demonstrate the high content variability achieved over different datasets, while still complying with shared source layouts. Indeed, the styles differ substantially between one sample to another, and variation is notably observed in local structures and elements, while still conforming to the layout at the global scale. This is most noticeable in the CelebA case (figure 18), where some of the images produced for given layouts depict young people while other old ones, and likewise some feature straight hair while other curly hair, even when originating from the same layout. Likewise, in the case of LSUN-Bedrooms (figure 17), we observe diversity that goes beyond aspects of texture and color scheme, featuring structural variations in entities such as windows, paintings, and bedding, among others. Meanwhile, figures 19-20 inversely show structural variability while maintaining a consistent style, obtained by rendering different layouts using the same set of style latents $\{w_i\}$.

The high diversity and consistency demonstrated here, which also quantitatively surpass the competing approaches (section 4.2), may be attributed to the new structural loss functions we employ (section 3.3). These purely generative losses liberate us from resorting to perceptual and feature-matching objectives common to prior work, which impose unnecessary and unjustified pixel-wise similarity conditions on the generated images, inhibiting their diversity. By sidestepping this reliance, we can unlock wider variation among the synthesized scenes, both in terms of structure and style. See a comparison between synthesized samples of different approaches at figures 22-24.

Table 4: **Paired vs. unpaired training**

|           | Paired | Unpaired | Parallel |
|-----------|--------|----------|----------|
| CLEVR     | **4.70**  | 6.58  | 6.72  |
| FFHQ      | **7.77**  | 8.12  | 8.35  |
| Cityscapes| **6.21**  | 7.32  | 7.81  |
| Bedrooms  | **6.05**  | 8.24  | 8.55  |
| COCO      | **21.58** | 25.02 | 27.41 |

Table 5: **Hyperparameters**

|           | Max #Latents | Latents Overall Dim | R1 reg weight ($\gamma$) |
|-----------|------|-----|-----|
| FFHQ      | 20   | 128 | 10  |
| CLEVR     | 16   | 512 | 40  |
| Cityscapes| 64   | 512 | 20  |
| Bedroom   | 64   | 512 | 100 |
| COCO      | 64   | 512 | 100 |

Table 6: **Dataset configurations**

| Dataset    | Size      | Resolution       | Augment    |
|------------|-----------|------------------|------------|
| FFHQ       | 70,000    | $256\times256$   | Flip       |
| CLEVR      | 100,015   | $256\times256$   | None       |
| Cityscapes | 24,998    | $256\times256$   | Flip       |
| Bedrooms   | 3,033,042 | $256\times256$   | None       |
| COCO       | 287,330   | $256\times256$   | Crop + Flip|

Table 7: **COCO subset statistics**

| Category   | Size  | Category   | Size  |
|------------|-------|------------|-------|
| **People** | **50293** | **Sports** | **37304** |
| Children   | 13840 | Skiing     | 5840  |
| Eating     | 5096  | Baseball   | 10067 |
| Playing    | 5798  | Tennis     | 11721 |
| Rural      | 12084 | Skating    | 6412  |
| Others     | 13475 | Surfing    | 3264  |
| **Animals**| **64066** | **Indoors** | **30946** |
| Dogs       | 5245  | Bathrooms  | 17121 |
| Cats       | 5543  | Kitchens   | 10427 |
| Birds      | 8153  | Bedrooms   | 3398  |
| Sheep      | 8134  |            |       |
| Bears      | 8138  | **Outdoors** | **20713** |
| Elephants  | 16204 | Beaches    | 8396  |
| Giraffes   | 6352  | Cities     | 8578  |
| Zebras     | 6297  | Streets    | 3739  |
| **Vehicles**| **53714** | **Misc** | **30294** |
| Airplanes  | 21232 | Desserts   | 11370 |
| Buses      | 7436  | Food       | 4176  |
| Trains     | 6661  | Toys       | 7242  |
| Bikes      | 18385 | Electronics| 7506  |

# C   Object Controllability & Amodal Completion

In section 4.4, we explore the model's spatial and semantic disentanglement, and study the degree of controllability achieved over individual objects and properties. Figure 13-16 provide a qualitative illustration, presenting examples of latent-space interpolations that lead to smooth and localized changes of chosen objects and properties, selectively controlling either their structure or style.

Thanks to the compositional nature of the layout generation, we can even add or remove objects from the scene, as is illustrated in figures 11-12, while respecting object interactions and dependencies such as shadows, reflections and occlusions. In particular, since GANformer2 creates the layouts sequentially by laying segments on top of each other (e.g. first generating a road, and then placing a car on top of it), it provides us with practical means to then remove the front segments and reveal the ones behind them, effectively achieving amodal completion of occluded objects. Consequently, by varying the number of generation steps, GANformer2 is also capable of extrapolating beyond the training data, e.g. creating empty CLEVR scenes (figure 3) even though the training data features at least 3 objects at every image.

# D   Structural Losses Comparison

As discussed in section 3.3, we introduce two new losses to the model's execution stage, where we transform input layouts into output photo-realistic images. The new losses of **Semantic Matching** and **Segment Fidelity** respectively encourage structural consistency between the layouts and the images, and fidelity at the level of the individual segment. In figure 21, we compare their performance in terms of FID score with several alternative objectives over the COCO dataset.

Specifically, we explore the following baselines: (1) Using *no consistency loss* at all (training with the standard fidelity loss only $\mathcal{L}(D(\cdot))$), in hopes that the model's layout-conditioned feature modulation will serve as an architectural bias to promote structural alignment; (2) A simple *concatenation* of the layout $S$ and image $X$ as they are fed into the discriminator $D$; (3) A *Feature-Marching loss*, as widely used in prior work, $\mathcal{L}_{FM}(f(X), f(X'))$, that compares VGG features of the generated image with those of the source natural image $X'$ that underlies the input layout $S$; and (4) an *Edge-Matching loss*, $\mathcal{L}_{EM}(e(S), e(S'))$, that compares the binary segmentation edges between the input layout $S$ and a layout $S'$ induced by the generated image $X$.

As figure 21 shows, our newly proposed losses, and the Semantic-Matching loss especially, surpass the discussed baselines and effectively encourage GANformer2 to generate high-quality images. For the Semantic-Matching loss, $\mathcal{L}_{SM}(S, S')$, we believe that providing semantic pixel-wise guidance to

| Layout | Pix2PixHD | BicycleGAN | SPADE | GANformer2 |
|---|---|---|---|---|

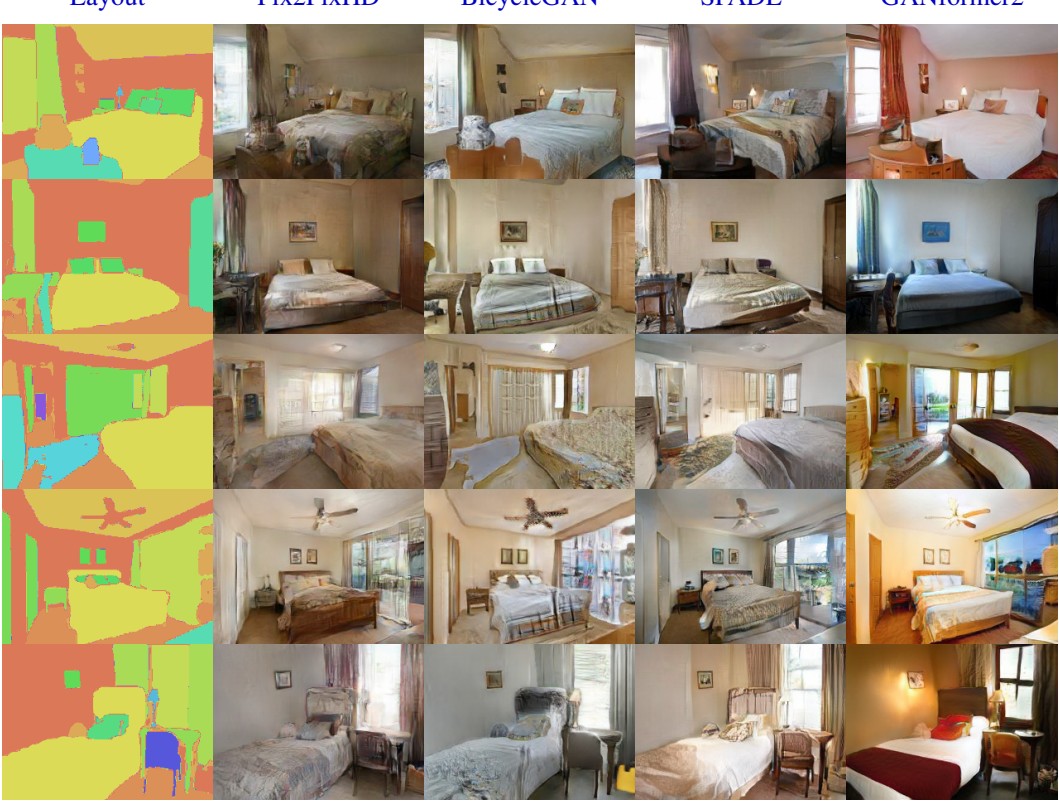

Figure 22: **Comparison between samples of conditional generative models**. GANformer2 achieves better visual quality and demonstrates a larger variance in the spectrum of colors and textures, contrasting with other approaches that converge to a narrow range of gray-brownish hues.

the generator in terms of the classes each region in the image should depict, while not limiting its content to match an arbitrary natural image, as is the case in feature-matching losses, accelerates the generator's learning without inhibiting its output diversity, which in turn yields better FID scores. Likewise, for the Segment-Fidelity loss, $\frac{1}{n}\sum \mathcal{L}_{SF}(D(s_i, x_i))$, promoting fidelity of individual segments $x_i$, rather than just of the whole picture $X$, naturally enhances learning, especially for rich and highly-structured scenes, as we observe for the COCO dataset.

## E    Model Ablations & Iterative Generation

To validate the efficacy of our approach and better assess the relative contribution of each design choice, we perform ablation and variation studies over the planning and execution stages. We begin by exploring the impact of varying the number of recurrent generation steps for planning the scene layout, and further compare it with a single-pass approach that generates the layout in a non-compositonal fashion – as a one image like standard GANs, rather then as a collection of interacting segments. We note that each generation step creates a random number of segments, sampled from a trainable normal distribution, and therefore reducing the number of steps does not limit the maximum number of segments the model can create overall. Adding more recurrent steps can instead enhance the model's capability to capture conditional dependencies across segments, while keeping the planning process shorter can naturally increase the computational efficiency.

**Compositional Layout Synthesis**. As figure 21 shows, compositional layout generation performs substantially better than the standard single-pass approach (denoted by "0") across all datasets. Intuitively, we believe that the efficiency gains arise from the ability of the recurrent approach to decompose the combinatorial space of possible scene layouts into several smaller tasks, such that each step focuses on a few segments only, rather than modeling the whole scene at once. This could be especially useful for highly-structured scenes with multiple objects and dependencies.

| Layout | Pix2PixHD | BicycleGAN | SPADE | GANformer2 |
|--------|-----------|------------|-------|------------|

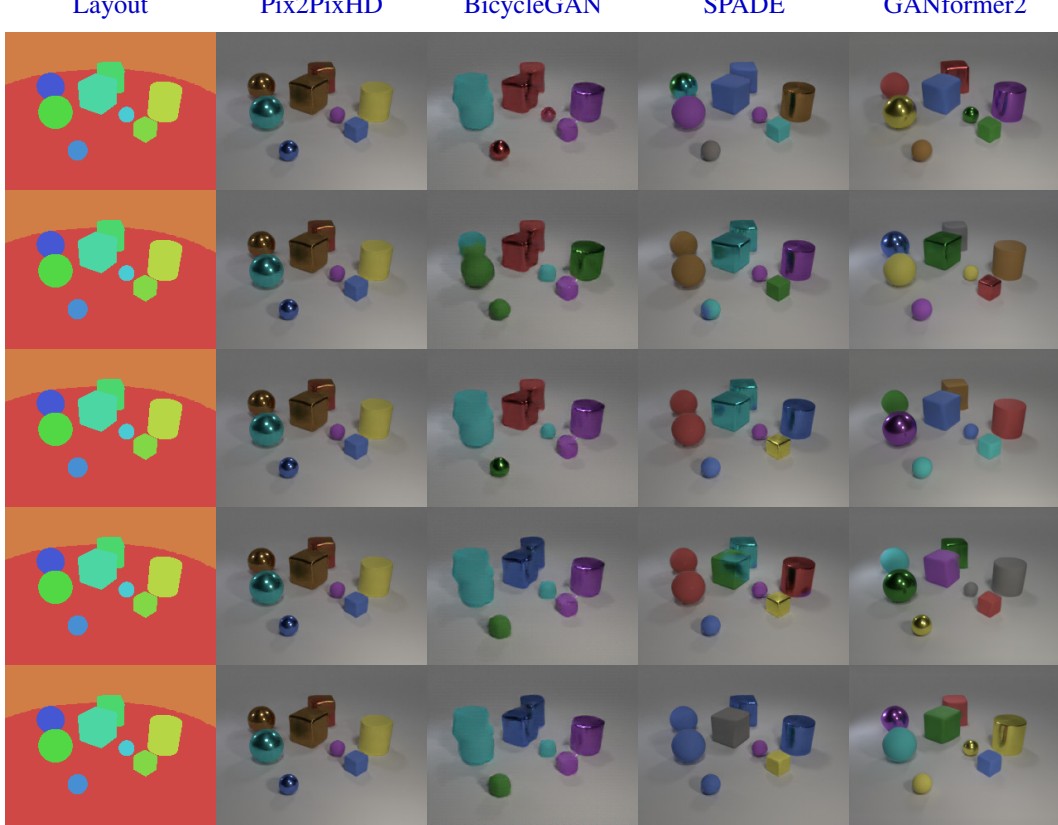

Figure 23: **Comparison between samples of conditional generative models**, demonstrating the GAN-former2's wider diversity in object's properties, its higher compliance with the source layouts, especially in terms of shape consistency, and the more precise separation between close object segments.

**Layout Synthesis Length**. We can see how the optimal number of synthesis steps vary between different datasets: while CelebA layouts seem simple enough for the model to comprehend and synthesize in just one step, CLEVR and LSUN-Bedrooms benefit from 2 steps, and the richer Cityscapes and COCO see performance improvement in even 3 layout generation steps. The most effective lengths seem to indicate the compositionality degree of each dataset: In CLEVR, objects are mostly independent of each other (holding only weak relations of not mutually occupying the same space), and so their segments can be produced mostly in parallel, over less recurrent steps. Meanwhile, more intricate scenes, as in COCO and Cityscapes, benefit from a longer sequential generation that can explicitly capture conditional dependencies among objects (e.g. generating a cup only after creating the table it is placed on), demonstrating the strength of recurrent synthesis.

**Layout Refinement**. In the execution stage, which transforms layouts into output images, we explore the contribution of the layout refinement mechanism (section 3.3). It introduces a sigmoidal gate $\sigma(g(S, X, W))$ to support local layout adjustments, meant to increase the model's flexibility and expressivity during the translation. We study ablations over CLEVR, either not applying the refinement or using limited gating versions, constraining the inputs to be the latents $W$, layout $S$, or image $X$ only. We see that compared to the default model's FID score of *4.70*, using weaker refinements lead to deterioration of 0.72, 0.78 and 0.85 when inputting the latents, layout or image respectively, and a larger reduction of 1.45 points, when ablating the gating mechanism completely. These results provide evidence for the benefit of using the gating mechanism to refine the scene's layout during the execution stage.

# F   Paired vs. Unpaired Training

We train GANformer2 over two sets: of images $\{X_i\}$ and layouts $\{S_i\}$, used during the planning and execution stages respectively. The training sets can either be paired: listing the alignment

| Layout | Pix2PixHD | BicycleGAN | SPADE | GANformer2 |

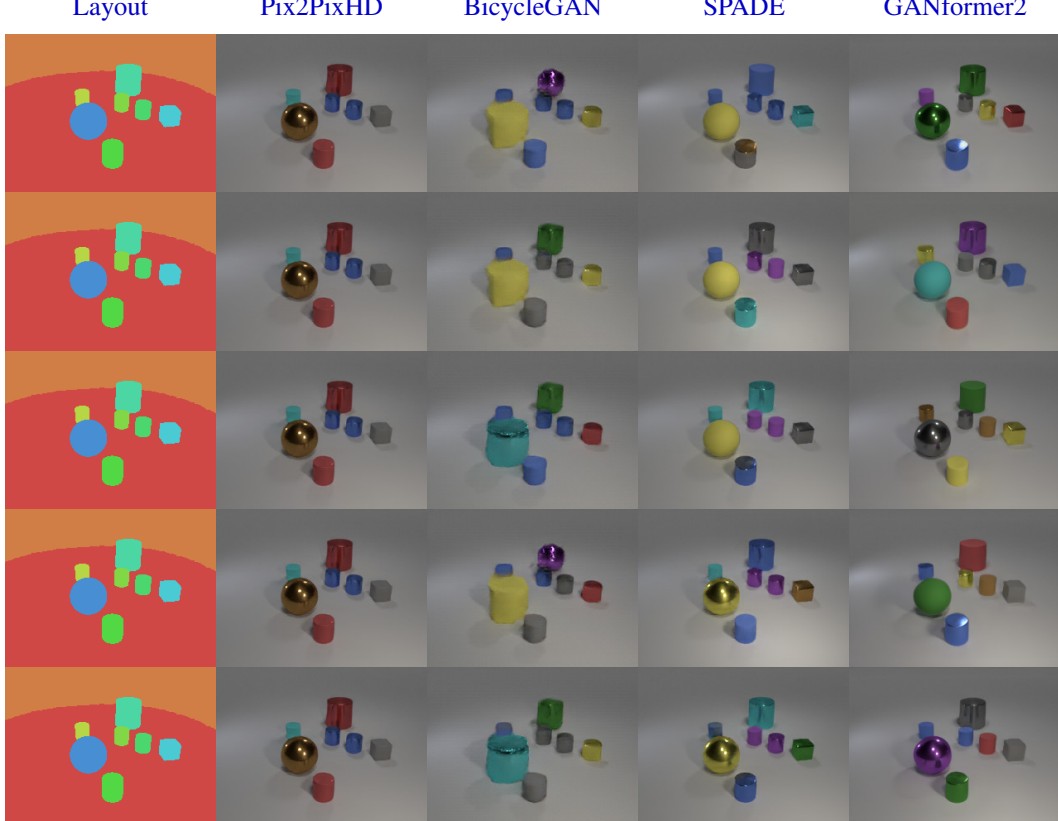

Figure 24: **Comparison between samples of conditional generative models**, demonstrating the GAN-former2's wider diversity in object's properties, its higher compliance with the source layouts, and the more precise separation between different object segments (***continued***).

between images and layouts, namely $\{(S_i, X_i)\}$, or unpaired: $\{X_i\}, \{S_i\}$, and our training scheme accommodates both options: For the paired case, we first train the planning stage over layouts $\{S_i\}$, and the execution stage conditionally over $\{(S_i, X_i)\}$, such that it learns to translate *ground-truth layouts* into output images. Then, to make both stages work in tandem, we fine-tune them together, where this time the execution stage translates *generated layouts* $\{S_i^g\}$ into output images $\{X_i^g\}$.

For the unpaired case, we can either (1) train the planning stage over the layouts $\{S_i\}$, and then fine-tune it together with the execution stage, or (2) jointly train the two stages from scratch (we call this scheme *Parallel*). Since the Segment-Fidelity loss assumes access to paired layout-image samples, both fake and real, we do not use it in the unpaired and parallel cases, and instead use the Edge-Matching loss, described in D. We adjust the Edge-Matching and Semantic-Matching losses for both the generator and the discriminator to be computed over generated pairs $\{(S_i^g, X_i^g)\}$, which are available even when the training data is unpaired. Table 4 compares the model performance between the paired, unpaired and parallel settings for different datasets. The he model manages to perform well even in the unpaired and parallel settings, but achieves strongest results in the paired case.

## G   Implementation & Training Details

We implement all the unconditional methods within the shared GANformer codebase [33], to ensures they are tested under comparable conditions in terms of training details, model sizes, and optimization scheme. For the conditional models, we use the authors' official implementations, likewise implemented as extensions to the Pix2PixHD repository for conditional generative modeling. All approaches have been trained with images of $256 \times 256$ resolution and data augmentation as detailed in section I. See table 5 for the particular settings and hyperparameters of each model. The overall latent dimension is chosen based on performance among $\{128, 256, 512\}$. The R1 regularization factor $\gamma$ is likewise chosen based on performance and training stability among $\{1, 10, 20, 40, 80, 100\}$.

In terms of the loss function, optimization and training configuration, we adopt the settings and techniques used in StyleGAN2 and GANformer [33, 42], including in particular style mixing, Xavier Initialization, stochastic variation, exponential moving average for weights, and a non-saturating logistic loss with lazy R1 regularization. We use Adam optimizer with batch size of 32 (4 times 8 using gradient accumulation), equalized learning rate of $0.001$, $\beta_1 = 0.9$ and $\beta_1 = 0.999$ as well as leaky ReLU activations with $\alpha = 0.2$, bilinear filtering in all up/downsampling layers and minibatch standard deviation layer at the end of the discriminator. The mapping layer of the generator consists of 8 layers, and ResNet connections are used throughout the model, for the mapping network, synthesis network and discriminator. All models have been trained for the same number of training steps, roughly spanning 10 days on 1 NVIDIA V100 GPU per model.

## H  Baselines & Prior Approaches

We compare GANformer2 to both unconditional and conditional generative models. First, we inspect unconditional methods which synthesize images from scratch, including in particular: (1) a baseline GAN [23], (2) the StyleGAN2 [42] model, (3) SAGAN [81] which utilizes self-attention across spatial regions, (4) k-GAN [69] that blends together $k$ generated images through alpha-composition, (5) VQGAN [21], a visual autoregessive autoencoder, (6) SBGAN [5], a non-compositional two-stage approach, and (7) the original GANformer [33].

We then proceed to compare our execution stage to popular conditional semantic generation models, including the aforementioned SBGAN and also: (8) Pix2PixHD [72] which uses a U-Net [62] to translate source to target images, (9) BicycleGAN [86] that promotes cycle consistency among domains, and (10) SPADE [60], which performs spatial modualtion using a fixed set of trainable semantic category vectors. For disentanglement and controllability experiments, we compare the GANformer2 to a baseline GAN, StyleGAN2 and GANformer, as well as: (11) MONet and (12) Iodine, two sequential variational autoencoders.

## I  Data Preparations

We train all models on images of $256 \times 256$ resolution, padded as necessary. See dataset statistics in table 6. The images in the Cityscapes and FFHQ datasets are mirror-augmented, while the images in the COCO dataset are both mirror-augmented and also randomly cropped, to increase the effective training set size. We assume access to a training data of image and panoptic segmentations [43], indicating the segment unique identity and its semantic class. Contrary to prior conditional works which rely on costly hand-annotated ground-truth segmentations, and to demonstrate the model robustness, we instead intentionally explore training on auto-predicted segmentations (either produced in an unsupervised manner [39] for CLEVR [37] or by pre-trained segmentor [74] otherwise).

For the COCO dataset [48], we note that it introduces challenges from two perspectives, being both **highly-structured**, with each scene populated by many objects that hold intricate relations and dependencies, but also visually and semantically **diverse**, consisting of varied images from a wide range of domains. In order to isolate the former challenge of modeling compositional scenes from the latter important but different challenge of covering a diverse image distribution, we study training on a topical partition of COCO, named $COCO_p$, that groups images into 7 semantically-related splits, listed in table 7. To partition the dataset, we cluster t-SNE processed ResNet activations of the COCO images into 31 subsets, which are then semantically grouped into the 7 splits. We train the models on each split separately, and report the mean scores. As expected, and also empirically suggested by table 1, the partition leads to improved visual quality across all models – both baselines and new ones, likely due to the more uniform resulting training distributions.