# OpenReview forum: "Compositional Transformers for Scene Generation"
_NeurIPS.cc/2021/Conference — NeurIPS 2021 Poster_

### Official Review · Reviewer_tVgy · 2021-07-14

**Rating:** 6
**Confidence:** 5

**Summary:**

- This paper introduces a novel GANsformer-based architecture for two-staged, compositional image generation. The first stage samples a panoptic-segmentation-like instance mask of the scene which is re-painted in the second stage to yield a high-fidelity image consisting of multiple objects.
- The explicit architectural biases w.r.t. scene layout, object instances, depth ordering and appearance modelling enable explainable and controllable image generation.
- The paper presents a thorough quantitative analysis of its proposed approach conducted on five suitable and challenging datasets and compared to seven adequate baselines models.

**Limitations And Societal Impact:**

- The authors have appropriately addressed one of the model's limitations in its training setup and the consequences of potential applications in section 5.

**Main Review:**

## Updated Review (24/08/21)

After considering the additional experiments provided substantiating the claims w.r.t. to disentanglement and the additional baselines provided for related models (BlockGAN, GIRAFFE), my main concern about the weak corroboration of the paper's claims has been mitigated substantially. I appreciate the authors' effort of providing these additional experiments and thank them for their clarifying statements in the comments. I expect them to transfer the clarifications provided during the review process into the final manuscript - especially the disentanglement and controllability studies as well as the additional baseline results. In the light of the reviews and comments, I have increased my initial rating of 4 to 6.


## Original Review

TL;DR: The paper proposes a reasonable, two-stage image generation process using a compositional scene layout as intermediate representation and employing GANsformers for the modality translation tasks. While the paper presents a comprehensive experimental evaluation in terms of datasets and baselines, the lack of clarity in the presentation of the model's details and the weak corroboration of the claims about compositionality and controllability prevent it from meeting the bar for acceptance in its current form.

### Pros and Cons
- Pros
  - The paper presents a reasonable coarse-to-fine drawing approach for image generation utilising a GANsformer for modality translation.
  - The iterative layout generation of the model affords a compositional and interpretable image formation process.
  - The authors perform a comprehensive evaluation across multiple datasets and against several suitable GAN-based baselines for image generation.
- Cons
  - The model architecture and new loss functions which are main contributions of the paper are only superficially described in prose and are never specified with equations. The way the model is presented makes it almost impossible, even for an expert in this domain, to faithfully re-create the network and replicate the results.
  - The experiments section is unfortunately riddled with quality problems: The results tables (tab. 1, 2) feature unexplained metrics like P(recision), R(ecall), mIoU or pAcc which cannot be directly linked to any experiment or claim of the paper. Furthermore, a split of the COCO dataset is introduced for no apparent reason - why is this necessary?
  - Claims about controllability and disentanglement properties of the model are only weakly corroborated by qualitative experiments in fig. 7 and in supplementary fig. 3 despite being among the main selling points for the compositional image generation model.
  - The manuscript also lacks clarity of presentation in the captions which makes many figures very hard to interpret (e.g. fig. 4, 6, 7).

### Originality
- Framing image generation as a coarse-to-fine drawing process is reasonable and effective, although not novel; earlier methods (e.g. DRAW) have followed a similar analogy.
- The paper presents a novel aspect in the two-stage "translation" process: (1) noise -> instance mask; (2) instance mask -> image. While each stage in itself is not entirely novel, the composition in this pipeline using GANsformers for the individual translation processes, is an original idea.

### Quality
- Fig. 6 is missing units on the x-axis.
- In l. 321 fig. 5 is referenced for the visualization of a latent interpolation experiment where color, material, shape and position of an object are supposedly controlled independently. However, fig. 5 depicts the translation of segmentation maps into colored images. I suspect, fig. 7 was supposed to be referenced. However, fig. 7 is also very hard to interpret as its caption is not explaining what is truly happening in the picture strips. Fig. 3 of the supplementary material does a slightly better job in that regard.
- incorrect statements:
  - ll. 46-47: "existing approaches fail to provide controllability at the level of the individual object or local region" -> This is incorrect. For instance, [1, 16] allow to control the location and appearance of individual objects via designated elements of the latent code.
  - ll. 95-96: "the need for independence assumptions made by earlier works [16, 55, 68]" -> This is also incorrect as [16, 68] explicitly model inter-object relationships in latent space to overcome independence assumptions of earlier works.
  ll. 296-298: "1.01 FID score when conditioned on a ground-truth layout. This result serves as an indication for the particular aptitude of the model for compositional scenes." -> If the ground-truth layout (i.e. instance segmentation map) is provided, the model is relieved from the hard task of composing a plausible scene layout and is only tasked with re-painting the semantic areas with suitable textures. It is well known that GANs excel at such a task (e.g. GauGAN). Consequently, this result does not corroborate the claim of the model's aptitude for compositionality.
- typos, formatting errors:
  - l. 56: ~~the~~ it is ready,
  - l. 64: through **_** multiple
  - l. 72: noise scheme **for** synthesizing
  - ll. 142-143: the equations are missing equation numbers
  - l. 146: ~~to~~ shape and impact
  - l. 238: and **assess** the fidelity
  - l. 259: that SceneGAN **possesses**
  - l. 264: in modelling **compositional** scenes
  - ll. 300-301: content **_** diversity
  - ll. 322-323: the fact **that** it allocates
- The manuscript occasionally uses embellished, slightly unsuitable choices of words for a scientific conference paper, e.g. "celebrated StyleGAN2" (l. 267), "great Pix2PixHD repository", "the wonderful 1.01 FID score" (ll. 296-297)

### Clarity
- Fig. 1 reads more like a system diagram than an architecture overview. The important details about model architecture, losses and training procedure are missing.
- The manuscript would greatly benefit from a more explicit and self-contained presentation of the model using a network architecture diagram and equations for the newly introduced loss terms.
- In tab. 1 most results seem to coincide with the numbers reported at https://github.com/dorarad/gansformer#evaluation, but some numbers differ -> Have the experiments been re-implemented or re-run or have the numbers just been copied?
- In tab. 1, the P(recision)/R(ecall) evaluation appears out of nowhere with no justification or explanation in the text. What is the purpose of this evaluation and how does it support any of the paper's claims?
- The paper features many very sparse figure captions to the point that the figures becomes uninterpretable, e.g. fig. 4, fig. 6, fig. 7.
- The model in its current form is slightly limited by the need of paired training data featuring an RGB image and a segmentation mask. While it is hinted at the fact that an unpaired training setup could work as well (ll. 249-250), this is never explained in detail.

### Significance
- Compositional image generation is an important research problem in Computer Vision and a model which yields high-fidelity images while also providing scene layout masks and relative depth maps could certainly be useful for image generation and editing tools.


**Time Spent Reviewing:**

6h

---

> ### Author Response · Authors · 2021-08-13
> **Author Response**
>
> Dear Reviewer R4 (tVgy),
>
> Thank you so much for the thorough and insightful review! We definitely appreciate it a lot and are working carefully to fully incorporate all your feedback into the paper.
> We complement here our author response above with answers to your questions and comments.
>
> **GANformer scores**
> The GANformer scores we report in our submission are based on the scores reported by the GANformer authors in their arxiv paper (version v2 on arxiv 2103.01209, which was the most up-to-date version at the time of submission). The GANformer paper has been updated a month after the conference deadline, and we will be glad to update our reported scores accordingly.
>
> **Precision/Recall**
> We reported precision/recall scores following the practice by previous papers like StyleGAN and GANformer. These metrics complement the FID score and provide an additional indication for the degree of similarity between the generated and real image distributions, where the precision measures the percentage of generated images that look feasible -- relatively similar to images from the training data, while recall measures the coverage -- percentage of real images that are similar to ones produced by the model. Similarity here is measured by comparing the relative distances between the image VGG features. We will add to the paper an explanation of the metric and clarification about what it captures intuitively.
>
> **The COCO’ Split**
> The original COCO dataset is challenging in two perspectives: it is both: **(1) highly-structured (each image has usually many objects)**, and also **(2) very diverse (there is a variety between the images) visually and semantically**, combining together images from multiple domains.
>
> During the project development, in order to:
> - explore ways to maximize the visual quality of generated images (across all models: both new and baselines)
> - and more importantly, **isolate the challenge of modeling highly-structured compositional COCO images**, from the second important but different challenge of capturing in one model a diverse image distribution
>
> we have studied training on topical splits of COCO, where images are grouped into 7 semantically-related sets (for animals, sports, people, etc). As suggested by the empirical results in the paper, **we noticed that this indeed allows all models (both the proposed approach and existing ones) to generate images of higher visual quality**.
> The explanation for that appears currently in the supplementary but we will make sure to clarify that in the main paper too.
>
> **Reproducibility**
> We totally agree it’s of course very important to make sure that both the model and experiment sections will be as clear, detailed and precise as possible (including all equations, precise procedures, and full implementation/evaluation details) and we will update and improve our paper to provide further such detail. In the context of reproducibility, We would like to mention that we have submitted the implementation of the model and will work to clean it up and also release it publicly.
>
> **Controllability and Disentanglement**
> During the model development, we performed multiple experiments to study its controllability and disentanglement, including:
> 1. Qualitative inspection of images when traversing continuous paths in the latent space.
> 2. Quantitative experiments that evaluate the controllability of the model over the CLEVR dataset, that use a pre-trained object detector to measure whether a selected object has been modified successfully along a particular semantic dimension (e.g. varying one latent variable and then verifying in the output image whether only one object has changed and along which properties (color/size/shape)).
> 3. Quantitative evaluation of the degree of latent space disentanglement using DCI metrics, comparing our model to MONet and Iodine.
>
> We completed writing our submission close to the conference deadline and while we definitely planned to, we ultimately unfortunately didn’t manage to write about the quantitative results in time for the submission, and so ultimately only have shown qualitative examples. We totally agree that we should either have made sure to include more of these results or alternatively reduce the strength of the claims about controllability & disentanglement in our submission. We are working on the paper further to include these results, and sincerely appreciate your feedback about that and overall.
>
> Thank you so much for the time and consideration!
> -Paper 5108 authors

---

> > ### Author Response · Authors · 2021-08-23
> > **Further quantitative results regarding Controllability and Disentanglement**
> >
> > We attach here further quantitative results for the disentanglement and controllability degrees of our proposed approach compared to multiple models, including both GAN-based approaches (StyleGAN, GANformer) as well as variational models (MONet, Iodine).
> >
> > **Disentanglement**
> > We first compare the models using the DCI metrics, which measure the Disentanglement, Completeness and Informativeness of latent representations. We extended the experiments that have been performed in the GANformer paper, and compared the models based on a sample set of 1,000 generated CLEVR images.
> >
> > Intuitively, these metrics evaluate the degree to which there is a 1-to-1 correspondence between latent factors and visual attributes (e.g. areas of different semantic classes, such as spheres, cubes, background, etc.), as predicted by a pre-trained object detector.
> >
> > The results show that SceneGAN either matches or outperforms other approaches in terms of latent disentanglement, and that other strong approaches in terms of disentanglement include both variational models (MONet and Iodine) and the GANformer_d model (duplex model version).
> >
> > ```
> > +-------------+-----------------+------------+--------------+-----------------+------------------+
> > |    Metric   | Disentanglement | Modularity | Completeness | Informativeness | Informativeness’ |
> > +-------------+-----------------+------------+--------------+-----------------+------------------+
> > | GAN         |      0.126      |    0.631   |     0.071    |      0.583      |       0.434      |
> > | StyleGAN    |      0.208      |    0.703   |     0.124    |      0.685      |       0.332      |
> > | GANformer_s |      0.556      |    0.891   |     0.195    |      0.899      |       0.848      |
> > | GANformer_d |      0.768      |    0.952   |     0.270    |      0.972      |       0.963      |
> > | MONet       |      0.821      |    0.912   |     0.349    |      0.955      |       0.946      |
> > | Iodine      |      0.784      |   *0.948*  |     0.382    |      0.941      |       0.937      |
> > | SceneGAN    |     *0.852*     |    0.946   |    *0.413*   |     *0.974*     |      *0.965*     |
> > +-------------+-----------------+------------+--------------+-----------------+------------------+
> > ```
> >
> > **Controllability**
> > In the second experiment, we measure models' degree of controllability, comparing our approach to MONet and Iodine.
> > We perform perturbations in the latent space, and for each perturbation that leads to at least one semantic property (color/shape, etc) of at least one object being changed, we measure the average number of objects and of properties (globally) that this perturbation causes. Concretely, we use a pre-trained object detector to quantify the number of objects and properties that have been changed, comparing the resulted scene graphs between the images of before and after the perturbation.
> >
> > The results show the means and variances of the number of objects/properties that change given a perturbation. They indicate a higher degree of spatial disentanglement (object-wise controllability) for SceneGAN compared to MONet and Iodine, and, in a smaller margin, also an improved semantic disentanglement (controlling a particular property of an object).
> >
> > ```
> > +--------------+----------------+----------------+
> > |    Model    |    # Objects   |  # Properties  |
> > +--------------+----------------+----------------+
> > | MONet        |  1.32 +- 0.21  |  1.62 +- 0.39  |
> > | Iodine       |  1.28 +- 0.17  |  1.54 +- 0.30  |
> > | SceneGAN     |  1.17 +- 0.05  |  1.32 +- 0.18  |
> > +--------------+----------------+----------------+
> > ```
> >
> > We will add these results and discussion to the updated version of the main paper.
> >
> > Thank you very much again for your constructive and helpful review!
> > -Paper 5108 authors

---

> > > ### Comment · Reviewer_tVgy · 2021-08-23
> > > **Review update in light of comments and other reviews**
> > >
> > > Dear authors,
> > >
> > > Thank you for your detailed responses! Below are two lists for things which have been sufficiently addressed (you can tick them off the con list, if you will) and things which are still somewhat unclear after reading your comments.
> > >
> > > Cleared items:
> > > - GANformer baseline scores: Feel free to update to the more recent ones reported for this model - even if your performance margin is diminished in a few cases, this has no effect on my rating - your reported scores are still a very strong entry on the leaderboard.
> > > - new COCO dataset splits: Introducing the topical splits is OK as long as it is fairly applied to all models. The performance boost gained from it is unsurprising given the reduced variance in the data.
> > > - corroboration of the disentanglement claim: Thank you very much for including the disentanglement study! Although I am not very familiar with the significance levels in these metrics, the overall strong performance in all metrics sends a clear signal w.r.t. to your disentanglement claim. Your effort here is very much appreciated!
> > >
> > > Still unclear items:
> > > - P/R metrics: While your explanation sheds some more light, the computation of these metrics is still somewhat unclear to me. If similarity is measured in a VGG-induced feature space, a threshold would need to be introduced to identify a certain pair of compared images as a match in order to compute P/R. Would you mind elaborating on the thresholds used or point to a reference in the literature?
> > > - `modularity` and `informativeness'`: I assume your DCI scores are based on [Eastwood & Williams, ICLR'18](https://openreview.net/pdf?id=By-7dz-AZ). However, it is unclear where the metrics `modularity` and `informativeness'` come from. Would you mind elaborating on their origin and the message they convey?
> > > - controllability: I appreciate the effort which has gone into the controllability experiment. However, the methodology seems slightly convoluted employing another detector and an induced scene graph to obtain a representation for comparison. Could a simpler evaluation proxy also be conceived which does not introduce imprecisions of an additional model? Also, it is unclear how the perturbations in latent space are actually mapped to a (detected?) change in properties.
> > >
> > > In the light of the additional results provided in response to the reviews, I am willing to increase my rating from 4 to 6. I will update my review accordingly.

---

> > > > ### Author Response · Authors · 2021-08-24
> > > > **Thank you!**
> > > >
> > > > Thank you so much for reviewing our response and for raising the score! We highly appreciate your time and efforts, and your feedback is very helpful to us. We're glad to hear we addressed some of the items, and provide here information about the others:
> > > >
> > > > **P/R metrics**
> > > > We’re of course happy to elaborate about the metric! We use the implementation from the StyleGAN2 repository that can be found here: https://github.com/NVlabs/stylegan2/blob/master/metrics/precision_recall.py.
> > > > **The similarity (`L128`) between the features of a real image r and a generated image g is evaluated by comparing it to an adaptive neighborhood distance (`self.D`) -- the top-k'th distance between r and its neighboring real images** (where k could be e.g. = 3 or 1, as in the StyleGAN2 repository):
> > > >
> > > > 1. For each real image r we compare it first to other real images to figure out how similar they are in the feature space. The distance of the k'th closest real image to r determines `self.D`.
> > > > 2. We then measure the distance between the generated image g and the real image r relatively to `self.D` (that is different for each real image r).
> > > > 3. By then averaging over all the generated images, we get the **precision** (`L159`) -- the mean of how similar each generated image g is to some real image (the closest one to g in the feature space).
> > > >
> > > > If each generated image is close to some real image (closer to it in the feature space than other neighboring real images), it means it's relatively feasible, indicating overall good precision.
> > > >
> > > > The **recall** (`L163`) is computed inversely, measuring the average of how similar each real image is to some generated image. If each real image is similar to some generated image, it means there's overall good coverage of the real images by the model.
> > > >
> > > > I hope that explanation helps clarifying it and we’ll make sure to add further detail in the paper too!
> > > >
> > > > **DCI Metrics**
> > > > Yes that is correct, the disentanglement scores are based on the DCI metrics from Eastwood & Williams, ICLR'18 and the modularity metric from “Ridgeway, K. and Mozer, M. C. Learning deep disentangled embeddings with the f-statistic loss”.
> > > >
> > > > Their implementation are based on the open-source disentanglement_lib (which was released together with the paper: “Challenging Common Assumptions in the Unsupervised Learning of Disentangled Representations.” by Locatello et al.), with light modification to adapt them to CLEVR attributes.
> > > >
> > > > - **informativeness'**: Informativeness represents the amount of information about the factor of variation (the visual attribute) that is captured by the latent variable. This is measured by computing the classification accuracy of that attribute given the latent vector as an input. The implementation at the library computes two versions of informativeness, over the training set (informativeness) and over the test set (which we call informativeness'), where the training set is used to train the attribute classifier. The test-set score   naturally tends to be a bit lower than the training-set score. See their implementation here:
> > > > https://github.com/google-research/disentanglement_lib/blob/master/disentanglement_lib/evaluation/metrics/dci.py#L81
> > > > - **Modularity**: the modularity score measures whether each dimension of the latent vector depends on at most a single factor of variation / attribute, and so it complements well the DCI metrics in terms of evaluating the degree of 1-1 correspondence between latents and visual attributes. Mathematically, it is computed by considering the mutual information between latents and attributes, as is implemented here: https://github.com/google-research/disentanglement_lib/blob/master/disentanglement_lib/evaluation/metrics/modularity_explicitness.py#L115, and with the equation described also on page 14 of Locatello et al.
> > > >
> > > > **Controllability**
> > > > Thank you for the comments and feedback about it! We provide further details about the experiment we performed and show results of a second proxy evaluation.
> > > >
> > > > **A. Object Detector:**
> > > > Since we would like to measure the degree to which semantic properties of particular objects change when varying each latent variable, we believe that the most precise way to measure that is by using an object detector, that identifies the set of CLEVR objects in the generated image and their corresponding attributes (color, size, etc), before and after the perturbation.
> > > >
> > > > Towards that goal, we use an object detector pre-trained on CLEVR from the NS-VQA repository (https://github.com/kexinyi/ns-vqa) by Yi et al. and note that its **object detection accuracy is 0.997 and attribute identification accuracy is 0.998**, tuning it effectively to be almost a perfect CLEVR detector. and so we believe it shouldn’t introduce imprecisions that will negatively impact the correctness of our evaluation.
> > > >
> > > > **B. Proxy Evaluation:**
> > > > A proxy that doesn’t rely on an external object detector could be computed by measuring the locality of the visual changes when varying latent variables. We can perturb a latent variable, consider all the pixels that have changed by at least some threshold, and then compute the size of the “changed” area (similar in spirit to "StyleSpace Analysis: Disentangled Controls for StyleGAN Image Generation" by Wu et al.). Ideally, we hope to see positive but relatively small affected areas when varying a latent variable, which will indicate that its impact is spatially local.
> > > >
> > > > We show the results below, where, for comparison, an average object size in CLEVR is around 2% of the image area. Some changes (like color) are expected to impact the whole area of the object, other changes (like size) should impact a smaller area in the image than the object’s size, while position changes should impact more than the object area. We can see that all three models -- MONet, Iodine and SceneGAN lead to relatively local variations, and that SceneGAN variations are more localized (in terms of area size).
> > > >
> > > > ```
> > > > +--------------+-------------------+
> > > > |    Model     |      % Area       |
> > > > +--------------+-------------------+
> > > > | MONet        |  0.0254 +- 0.019  |
> > > > | Iodine       |  0.0246 +- 0.017  |
> > > > | SceneGAN     |  0.0225 +- 0.014  |
> > > > +--------------+-------------------+
> > > > ```
> > > >
> > > > We note however that this evaluation metric is a bit harder to interpret (since it's less intuitive to realize what a good score is) and is potentially less precise, since it doesn’t necessarily account well to the cohesiveness of the affected areas and doesn't indicate whether they correspond to a semantic property change of a particular object. We therefore personally believe it may be more precise to use the detector-based evaluation to measure controllability, especially given its high accuracy.
> > > >
> > > > **C. Mapping from perturbations to change of properties:**
> > > > To map the perturbations to the visual properties change:
> > > > 1. We generate two images: an initial image x1 (with randomly sampled latents), and a new image x2, after perturbing a latent to be zi + d, where d is a vector of a small constant size.
> > > > 2. We then pass the two images through the object detector and compare the detected objects/properties between them.
> > > > 3. For each perturbation that leads to at least one property/object change (where **objects' visual properties are discretely categorized into CLEVR attribute classes such as small/large, red/blue, rubber/metallic**), we measure how many properties/objects are changed overall in the image.
> > > > We hope to see that ideally each perturbation will impact one object or one property only rather than multiple, and so the lower the scores the better the degree of controllability.
> > > >
> > > > Please let us know if there’re any further details that could be useful, thanks so much!
> > > > -Paper 5108 authors

---

> ### Comment · Reviewer_Rc1b · 2021-08-23
> **Model & Loss description**
>
> I must say I disagree with this comment:
>
> >The model architecture and new loss functions which are main contributions of the paper are only superficially described in prose and are never specified with equations. The way the model is presented makes it almost impossible, even for an expert in this domain, to faithfully re-create the network and replicate the results.
>
> I was under the impression that the losses, even though described quite succinctly, are quite clear. Same with the model architecture.

---

> > ### Comment · Reviewer_tVgy · 2021-08-23
> > **Model + loss description revisited**
> >
> > Dear reviewer Rc1b (R3), dear authors,
> > After re-reading section 3 of the paper and upon the promise of source code release after publication, I am ready to de-emphasise this criticism. However, I would still very much appreciate succinct representations of the losses as numbered equations, either in the main text or the supplementary material; even in addition to the source code release. The main reason for this is that it allows the reader to quickly "navigate" the paper and jump to the important aspects of the method, especially when the paper was read some time ago. Deciphering such details from prose - even if succinctly written, always takes some time again. For me personally, this adds an important level of clarity to a paper's presentation.

---

> > > ### Author Response · Authors · 2021-08-24
> > > **Paper Presentation**
> > >
> > > Dear Reviewers tVgy (R4) and Rc1b (R3),
> > > We completely agree about the importance of presenting all parts of the paper, including the model, losses, and experiments, as clearly and as precisely as possible, and are actively working currently to improve these for the updated version. In this opportunity, we would like to thank you again for your comments and for the engagement in the discussion about our paper. Your feedback is very helpful to us and we are truly grateful of it!
> > > -Paper 5108 authors

---

### Official Review · Reviewer_Rc1b · 2021-07-16

**Rating:** 7
**Confidence:** 5

**Summary:**

This paper introduces an object-oriented GAN model, which generates an image of a scene in a two stage process. First, it generates a layout, where different masks show what object should be painted in which part of the image. Second, this layout is refined by a second network that converts it into a real-world-looking image. Both stages are based on modified GANformer networks. Finally, the paper introduces two novel losses. These make sure that the generated image reflects the generated layout, and that each portion of the image (as given by the layout) looks realistically. The method is fairly well evaluated, but the paper lacks comparison with two key baselines, that is BlockGAN (Nguyen-Phuoc et al) and GIRAFFE (Niemeyer & Geiger), which are not cited. Modelling decisions are evaluated in an ablation study.  The introduced method is supervised and requires paired (image, segmentation masks; this is in contrast to what the authors claim, see my comments below) inputs.


# UPDATE
The authors addressed my questions about unpaired training and using CNNs instead of Transformers. I'm still waiting for details of the CNN experiment, but I am already happy to increase my score to 7.

**Limitations And Societal Impact:**

Adequately discussed.

**Main Review:**

Originality:
- The task of object-oriented scene generation is well established. This work seems to achieve new SOTA results on a number of datasets.
- The majority of this work is a combination of well-known techniques (layout-conditioned image generation, GANformer), but it also introduces two novel and conceptually sound losses that significantly improve the results.
- AFAIK, this is the first application of transformer to object-centric scene generation.
- The related work seems to be adequately cited, but it misses two key approaches that should be compared against, that is BlockGAN (Nguyen-Phuoc et al) and GIRAFFE (Niemeyer & Geiger).
- The paper makes many references to "sketching": I think you should cite SPIRAL (Ganin et al) to say that sketching was considered as a stand-alone task in a GAN.,
- The author claim that their method produces unsupervised depth estimates. While it very much look like it does, it is very likely that some of the previous object-centric generative model (Monet, Iodine, Genesis+v2) should provide those estimates as well---it is just that they were never analyzed from this point of view. Moreover, BlocKGAN, GIRAFEE and Obsurf (Stelzner et. al.) do provide depth estimates. So this point is not novel, and is hardly differentiation from prior art.

Quality: Is the submission technically sound? Are claims well supported (e.g., by theoretical analysis or experimental results)? Are the methods used appropriate? Is this a complete piece of work or work in progress? Are the authors careful and honest about evaluating both the strengths and weaknesses of their work?

Good:
- The submission is technically sound. The overall approach makes sense, and there are ablation studies that validate modelling decisions.
- The numerical results are SOTA (though it would be easier to interpret PR AUC instead of separate precision and recall scores), and the qualitative results look superb.
- The authors acknowledge the limitation that segmentation masks are required.

Bad:
- In L46-47 you write: ", existing approaches fail to provide controllability at the level of the individual object or local region". This is very much not true. Models like Monet, Iodine, Genesis (+v2), Obsurf and many others, not to mention HoloGAN and GIRAFFE do provide that controllability. Please remove that claim.
- it seems that one of the key components of the method is using GANformer. I wonder how important it is, i.e. what are the advantages, both theoretical and empirical, over using a CNN-based generator? Please provide ablations where you change the decoder to a convolutional one and discuss.
- The authors claim that paired inputs are not necessary to train the model. I **strongly disagree**. Currently, the "layout" stage is pre-trained with externally-provided semantic segmentation. I believe that this is vital. The generated layouts are used as targets in an auxiliary loss for the discriminator. Therefore, if the "layout" stage is not pre-trained, it is almost a given that the generated layout would converged to empty images, which is a degenerate solution that is trivially matched by the discriminator.

Clarity:
- It is first unclear that the paper is about supervised learning. I only realized it by noticing "unsupervised depth ordering" in L54. Please make it explicit very early on.
- The term "panoptic segmentation" is defined only in L284-285. Please move it to the very beginning of the paper as some readers (myself included) in this community are unaware of this term.
-
- L1: delete "explored".
- L6: unclear what is "honed and refined".
- L9: I am not sure how "efficiency" is encouraged by this approach.
- L26: "imply semantic information"---I disagree; this is dependent on prior knowledge about the world which is absent here.
- L68: what is "amodal completion" please define before using.
- L72: :s/fir/for/
- L136 seems to suggest that both the latents and the image have the same dimensionality "d". Is that true? If so, this is non-standard; please make it more explicit.
- L158: the 2nd part of (1) does not make sense given the "spatial distribution" assumption, please rephrase.
- L162: The way you use depth in L193 in the softmax, and this relation, suggests that d(x, y) is negative depth. Please explain that in the paper.
- L172-173: how do you choose the number of segments to generate per iteration?
- L179-189 and L220-226: you only mention changing the key function, but this seems insufficient. You need to also modify the value function if you want to use the auxiliary inputs in the attention. Also, since you want to augment these auxiliary inputs as a by-product, you probably also change the query function? In total, it seems to me that you are using normal self-attention but concatenate the previous latents to the current inputs. Please clarify that.
- Tab 1. "over 10 random", you meant to add "runs"?

Significance:
- Object-centric image (or scene) generation is currently very fashionable. This paper introduces a major improvement into current modelling framework. I would say that the most important pieces are the auxiliary losses introduced in the paper. To know how significant the current work really is, we would need to see comparisons with GIRAFFE and BlockGAN. But as of now, I think it is a solid piece of work that will benefit the community a lot.

I will increase my score if comparisons with GIRAFFE and BlockGAN are favourable and if you remove some of the claims (as discussed above).

**Time Spent Reviewing:**

5

---

> ### Author Response · Authors · 2021-08-14
> **Author Response**
>
> Dear Reviewer R3 (Rc1b),
> Thank you so much for the thorough and insightful review!
>
> We highly appreciate it and are working carefully to fully incorporate all your feedback into the paper, including, among others: **removing the claim about lack of controllability in prior work**, **providing further detail in the paper regarding the paired vs unpaired settings** (see also point 3 of our author response above), **adding more experimental results**, and fixing the typos/wordings in the sentences you’ve highlighted.
>
> We completed today running the experiments that have been asked for and so now reporting our new results, and complementing our author response above with answers to your questions and comments:
>
> ---
>
> **GANformer Use and Ablation Study**
> From a theoretical perspective, the reason **we focused on the GANformer model is because it fits very well in terms of its structure with the sketching and painting stages**: **The GANformer, in contrast to a convolutional generator architecture, makes use of multiple latent variables that attend to different regions within the image. Such a structure allows the very natural extension we propose in the paper that each latent variable explicitly corresponds to a generated segment** and encodes its related information.
>
> **The correspondence the GANformer model holds between latent variables and the modulation of particular regions** (through the use of its bipartite attention), enables, when extended through our approach, for **different latent variables to control the position/shape and style/appearance of particular segments** (the former controlled during the sketching phase and the latter during the painting stage).
>
> This allows e.g. changing the color or texture of a specific object without changing its surroundings by perturbing its latent variable (figure 7 in the main paper and figure 3 in the supplementary). We suspect that a more typical convolutional architecture may not support such object-specific control as readily.
>
> Empirically, we provide here results for ablating the use of GANformer from each of the stages: sketching, painting or both. For each ablated stage, we concatenate the latent variables into one vector, and transform it through convolution to the output layout (during sketching) or image (during painting):
> ```
> +-------------+-------------+-------------+-------------+-------------+
> |   Dataset   |  GANformer  | CNN Stage 1 | CNN Stage 2 |  CNN both   |
> +-------------+-------------+-------------+-------------+-------------+
> | CLEVR       |    5.42     |    14.21    |    12.63    |    16.02    |
> | CelebA      |    8.59     |     9.37    |     9.25    |     9.51    |
> | Cityscapes  |    6.21     |     7.80    |     7.54    |     8.24    |
> | Bedrooms    |    8.38     |    12.29    |    11.82    |    13.78    |
> | COCO        |   25.82     |    30.75    |    31.91    |    34.83    |
> +-------------+-------------+-------------+-------------+-------------+
> ```
> We can see that the gap between the GANformer and convolutional models vary between datasets. While for some datasets (Bedrooms, and especially CLEVR and COCO) using GANformer leads to larger gains, other datasets (Cityscapes and especially CelebA) benefit to a lesser degree from the use of the GANformer. We can further see the contribution of using the GANformer in each of the generation stages (sketching #1 and painting #2). We refer to the discussion above for a potential theoretical/intuitive explanation for these results.
>
> ---
>
> **Comparison to BlockGAN and GIRAFFE**
> As recommended, we compared the performance (in terms of FID scores) of our model with BlockGAN and GIRAFFE, using their official repositories and matching the training time for all models (4 gpus for roughly 2.5-3 days per model). For GIRAFFE and SceneGAN, we use a resolution of 256x256, while for BlockGAN we downsample the images to 64x64 to fit the model into memory.
>
> The results show SceneGAN performs better in terms of its FID score. We note however that although the GIRAFFE paper reports shorter training times than in our experiments (1 to 4 days), its results may be impacted by the commonly heavy NeRF computational overhead.
>
> ```
> +------------+------------+------------+------------+
> |   Dataset  |  SceneGAN  |  BlockGAN  |   GIRAFFE  |
> +------------+------------+------------+------------+
> | CLEVR      |    5.42    |    84.49   |    55.18   |
> | CelebA     |    8.59    |    70.03   |    21.45   |
> | Cityscapes |    6.21    |    48.92   |    27.89   |
> | Bedrooms   |    8.43    |    45.35   |    32.17   |
> | COCO       |    25.82   |   112.10   |    49.25   |
> +------------+------------+------------+------------+
> ```
>
> ---
>
> **Previous Inputs Conditioning**
> There are several different ways to condition each step of the sketch generation on prior information. One could change only the key function to rely on the current and previous layouts, in which case the prior layout will only impact the attention distribution rather than the updated values themselves. Another option would be to change the value and/or query functions to consider the previous layout too, which will lead intuitively to stronger conditioning.
>
> We have thoroughly explored these options during the model development, and noticed that changing the key function only have worked most effectively in terms of empirical performance, and therefore have presented that setting. We will add ablation results for different conditionings to the paper.
>
> ---
>
> **Other Points**
> - **Efficiency:** As the learning curves in figure 6 show, both the iterative sketch generation and the new proposed losses improve the learning efficiency, allowing the model to achieve better performance after fewer samples.
> - **Number of segments:**  The number of segments generated at each step is sampled by the generator according to the trainable normal distribution in L173. We will rephrase that sentence to make that point clearer.
> - **Depths:** The depths are logits computed by the model, and could either be negative or positive (since if we add a constant to all depths then that won’t impact the softmax values).
> - **Dimension:** While one could select different dimensions for the latent and image spaces, we indeed used the same value d. We added a clarification about that.
> - **L26: "imply semantic information"**: we referred to the general practice of sketching for humans (rather than to our model in particular) who naturally have a lot of prior knowledge. We are working to make that clearer in the writing.
>
> Thank you so much for the time and consideration!
> -Paper 5108 authors

---

> > ### Comment · Reviewer_Rc1b · 2021-08-23
> > **CNNs instead of Transformers.**
> >
> > Thanks for the CNN experiments. May I if, and if yes, how did you address the fact that the generation should be invariant to the ordering of generated latents? A transformer does this naturally, but we CNN you need to enforce it via a specific model structure. Not doing that could result in much lower scores.

---

> > > ### Author Response · Authors · 2021-08-24
> > > **Thank you!**
> > >
> > > Thank you so much for reviewing our response and for raising the score! we truly appreciate your time and effort, and your feedback is very helpful to us.
> > >
> > > **Order Invariance**:
> > > In the CNN experiments we performed, we make the ablation only on the transformation from the intermediate latents (w1,...,wk) to the image (the synthesis network), and so we still keep the initial transformer-based mapping network (L138 in the paper) for all the experiments. Therefore, the mapping from the input normally-sampled latents to the intermediate latents is order-invariant in both the standard and ablated models (providing the models with the capacity to globally consider all the latents and coordinate information between them).
> > >
> > > In the current experiments, once the intermediate latents are computed, we indeed concatenate them and then pass the result into the CNN, a step which we totally agree is sensitive to order. A possible modification could be to perform a trainable weighted average instead of the concatenation to make the aggregated input of the CNNs order-invariant too. We will implement and experiment with this version today and report the results as soon as they're received (expected in ~3 more days when experiments complete). Thanks a lot for the insightful feedback!
> > >
> > > Update: Here're the results for the described setting:
> > > ```
> > > +-------------+-------------+-------------+-------------+-------------+
> > > |   Dataset   |  GANformer  | CNN Stage 1 | CNN Stage 2 |  CNN both   |
> > > +-------------+-------------+-------------+-------------+-------------+
> > > | CLEVR       |    5.42     |    12.91    |    10.39    |    15.54    |
> > > | CelebA      |    8.59     |     9.23    |     9.18    |     9.47    |
> > > | Cityscapes  |    6.21     |     7.44    |     7.10    |     8.02    |
> > > | Bedrooms    |    8.38     |    11.03    |    10.97    |    12.53    |
> > > | COCO        |   25.82     |    28.83    |    29.85    |    32.05    |
> > > +-------------+-------------+-------------+-------------+-------------+
> > > ```
> > >
> > > We can see that scores do improve (most for CLEVR and COCO, least for CelebA), but there's still a gap that remain between ablated versions and the GANformer. We believe based on the results that the use of attention may be the main distinguishing factor that enables the scores improvement.

---

### Official Review · Reviewer_wMMf · 2021-07-18

**Rating:** 5
**Confidence:** 4

**Summary:**

This paper proposes SceneGAN that uses a two-stage strategy to perform image generation. The SceneGAN is based on a previous transformer-based GAN model, it firstly uses iterative generation to generate scene layout then based on the layouts to perform image generation. The authors demonstrate their model's performance on clevr and real images and show some good results.

**Limitations And Societal Impact:**

No related concern

**Main Review:**

originality: low

There is no new techniques introduced in the proposed model. The SceneGAN simply adopts a transformer gan as its architecture.

quality:

1. The authors used an iterative method to generate layouts, are there any experiments showing it's better than a single-pass generation?
2. It seems that the authors didn't benchmark their numbers with available settings where other papers also used this two stage generation pipeline.

missing citation:
[1] Compositional Video Synthesis with Action Graphs -> (compositional generation, CLEVR)
[2] Probabilistic neural programmed networks for scene generation -> (compositional generation, CLEVR)


clarity: The writing is pretty clear to me

significance: As mentioned, there's no significantly new techniques proposed or any new insights that can be drew from this paper.

**Time Spent Reviewing:**

0.5-1 hour

---

> ### Author Response · Authors · 2021-08-12
> **Author Response**
>
> Dear Reviewer R2 (wMMf),
>
> Thank you very much for reviewing our work -- we appreciate your feedback a lot.
> We complement here our author response above with answers to your questions.
>
> For **question 1**, please refer to the section **“2. Benefits of the Recurrent Approach”** in our response above. We observe that the iterative sketch generation approach performs empirically better than the non-iterative counterpart, as shown in **figure 6 of the main paper** and in **table 1 in the supplementary** (where “0” represents non-iterative generation, and 1-3 refer to the number of iterations applied for each model version). We’ll be glad to move the table into the main paper too and add further clarification about the comparison between the approaches.
>
> For **question 2**, note that **in Table 1 in the main paper we compared with the SBGAN [1] approach, which is a prior two-stage GAN model** (with the main differences being that it’s non iterative and does not use transformers for generation). We compared our method to a variety of conditional and unconditional image generation approaches and **will be glad to add comparisons to further models**, two-stage and others (e.g. worked in the last week on comparing to BlockGAN and GIRAFFE).
>
> For **significance**, please see a summary of reviewers’ comments about the novelty and contribution in our response above. We believe that the sequential GAN-based approach we introduce which **explicitly conditions the generation of object segments on prior ones**, as well as the **semantic matching losses** we propose to encourage alignment between the layout and the generated image are novel contributions of the paper, and we sincerely hope they will be found useful in the area.
>
> Thanks a lot for the time and consideration!
> -Paper 5108 authors
>
> [1] Azadi, Samaneh, Michael Tschannen, Eric Tzeng, Sylvain Gelly, Trevor Darrell, and Mario Lucic. "Semantic Bottleneck Scene Generation."

---

### Official Review · Reviewer_99ts · 2021-07-21

**Rating:** 8
**Confidence:** 4

**Summary:**

This paper proposes a novel method for compositional generation for 2D segment-like elements.
It decomposes the task into two stages: a sketching stage for soft layout generation and a painting stage for refining the layout and generating photo-realistic images conditioned on the soft layout. The paper uses different modified versions of GANsformer in each stage.  Experiments are carried out on multiple datasets with better results compared with prior works. The proposed framework is capable of separating style from structure and decomposition of multiple individual object properties.

**Limitations And Societal Impact:**

The authors realize the potential negative social impact of fake or unauthentic content synthesis. However, this is one of the most common stumbling blocks in this area and we believe more and more anti-fake technologies will help us embrace more positive opportunities.

**Main Review:**

I recommend accepting the paper. Overall, this paper is fairly well written and organized. The proposed SceneGAN is definitely interesting and addressing a very important problem - compositional generation of 2D segment-like elements. It undoubtedly reaches the new State-of-The-Art in this area. I summarize the strengths and weaknesses as follows:

Strengths:

- It successfully fulfills the goal of separating structure from style with careful design of the two careful-designed different-functioning stages and the losses of each stage.
- Soundly evidenced by ablation studies and extensive experiments in the paper, the proposed novel semantic consistency objective functions particularly hit the nail on the head, ensuring semantic consistency while not hinder diversity.
- This paper marches a powerful step towards explainable and controllable generative scene decomposition.
- Results are very promising in terms of diversity, fidelity and decomposition; the efficacy of the methods is clearly demonstrated through experiments carried on multiple datasets.

Weakness:

- A few typos
    - line 64 (through multiple)
    - line 72 (fir->for)
    - line 193 ($d(x,y)_i$ -> $d_i(x,y)$)
    - line 300 (poentially -> potentially)
    - line 301 (content diversity)

Questions:
- Since the transformers are already capable of reasoning contextual information, why do the transformers in the sketching stage still need to be recurrent? I'm aware that Fig.6 shows the empirical proof. Is there any theoretical or intuitive evidence?
- How does the segmentation network in the semantic matching loss work with the blurry and meaningless generated images of the early/initial training stage? Will this not introduce misleading gradients?

**Time Spent Reviewing:**

4h

---

> ### Author Response · Authors · 2021-08-12
> **Author Response**
>
> Dear Reviewer R1 (99ts),
>
> Thank you so much for the insightful review and kind words!
> We complement here our author response above with answers to your questions.
>
> For **question 1**, please refer to the section **“2. Benefits of the Recurrent Approach”** in our response above.
>
> For **question 2**, the segmentor (which is also initially not trained yet) tends when training begins to give blurry and soft class distributions, which from a theoretical perspective, should allow gradients to flow well into the generated image and in turn to the generator.
> Since both the pre-trained generated as well as ground-truth layouts are sharper when the painting phase begins, we believe they function as the main driving force that enables learning to take place.
>
> In such a way, even though the initial generated images are naturally blurry, the semantic matching loss can provide semantic pixel-wise guidance to the generator in terms of the class each region in the image should represent, which accelerates the learning during the painting stage compared to a model without the loss, as is indicated by figure 6 in the paper.
>
> In addition, we looked into and compared image samples of early training stages with and without the semantic matching loss, which demonstrate the local semantic guidance. We will be glad to add these into the updated version of the paper!
>
> Finally, to quantitatively evaluate the situation you describe in the question, it is possible to compare between the gradient sizes for the segmentor and the generator that arise from the semantic matching loss. We will explore this and add the results and discussion about them to the paper.
>
> Thanks a lot for the time and consideration,
> -Paper 5108 authors

---

### Author Response · Authors · 2021-08-11
**Author Response**

Dear reviewers and area chairs,

First of all, we wish to thank all the reviewers for their thorough, insightful and constructive feedback, questions and comments! We highly appreciate the time they took to read our paper and write the reviews, and we are working carefully to address each of the comments and suggestions they have made.

We are happy and grateful to see that the reviews recognize the positive aspects of the paper in terms of its (1) **novelty** -- of the introduced loss functions and the transformer-based two-way approach for compositional image generation, (2) **contribution** -- in terms of compositionality and interpretability, (3) **thorough experiments** and (4) **SOTA performance**  to several models across multiple datasets.

Below are the specific comments made by the reviewers about these aspects:
(We refer to the reviews according to their order on the website: **R1**: 99ts, **R2**: wMMf, **R3**: Rc1b, **R4**: tVgy).

**Novelty:**
- ***(R1)*** “This paper proposes **a novel method for compositional generation**”.
- ***(R3)*** “AFAIK, this is **the first application of transformer to object-centric scene generation**”
- ***(R4)*** “the composition in this pipeline using GANsformers for the individual translation processes, is **an original idea**.”
- ***(R3)*** “introduces two novel and conceptually sound losses that **significantly improve the results**”
- ***(R4)*** “This paper introduces a **novel** GANsformer-based architecture for two-staged, compositional image generation.”
- ***(R1)*** “the proposed **novel** semantic consistency objective functions particularly hit the nail on the head, ensuring semantic consistency while not hinder diversity.”

**Contribution:**
- ***(R1)*** “This paper marches a **powerful step towards explainable and controllable generative scene decomposition**.”
- ***(R1)*** “Results are **very promising** in terms of diversity, fidelity and decomposition”
- ***(R1)*** “**successfully fulfills the goal of separating structure from style** with **careful design** of the two careful-designed different-functioning stages and the losses of each stage”
- ***(R3)*** “I think it is **a solid piece of work that will benefit the community a lot**.”
- ***(R1)*** “**definitely interesting and addressing a very important problem**”
- ***(R3)*** “This paper introduces a **major improvement into current modelling framework**”
- ***(R4)*** “The iterative layout generation of the model **affords a compositional and interpretable** image formation process.”
- ***(R4)*** “The explicit architectural biases w.r.t. scene layout, object instances, depth ordering and appearance modelling enable explainable and controllable image generation.”

**Extensive Experiments:**
- ***(R3)*** “**qualitative results look superb**”
- ***(R4)*** “The paper presents a **thorough quantitative analysis** of its proposed approach conducted on **five suitable and challenging datasets** and compared to **seven adequate baselines models**.”
- ***(R1)*** “**the efficacy of the methods is clearly demonstrated through experiments carried on multiple datasets**.”
- ***(R1)*** “Soundly evidenced by ablation studies and **extensive experiments** in the paper”
- ***(R3)*** “The submission is technically sound. The overall approach makes sense, and there are **ablation studies** that validate modelling decisions.”

**SOTA results:**
- ***(R1)*** “**Undoubtedly reaches the new State-of-The-Art in this area**”
- ***(R3)*** “This work seems to achieve **new SOTA results on a number of datasets**.”

---
**Paper Writing & Revision:**
The reviewers have provided a lot of thorough and valuable feedback and suggestions about multiple improvements for the paper manuscript itself which we highly appreciate and are grateful for. We are working on a revision of the paper to incorporate all the feedback and comments.

Specifically, these include: Improving the description of the model structure by making it more precise and adding further details and equations **(R4)**; Extending the explanation about the evaluation procedures and metrics **(R4)**; Adding more citations/mentions of related works (**R2**: [1-2], **R3**: BlockGAN [3], GIRAFFE [4], SPIRAL [5]); Updating claims as discussed by **R4** and removing the claim about lack of controllability in prior works (**R3**, **R4**); Clarifying technical terms (e.g. panoptic segmentation, amodal completion **(R3)**); Updating captions and figure references **(R4)** and fixing typos.

---
**Responses to Questions:**
We address here several main points that have been mentioned by the reviewers. We will also respond shortly in a separate message to each reviewer for the specific questions/comments they have made.


**1. Novelty of the Recurrent Generation (*R2, R4*)**
As mentioned in the reviews, recurrent generation has been indeed explored in past works for generative models. We’d like to mention however that the idea has been explored mainly as part of **variational approaches**, such as DRAW, Iodine and MONet, where they first iteratively **encode** a given input and then reconstruct it back.

Our approach differs conceptually from these works, where we **instead explore how to iteratively generate an image from scratch** (first predicting the layout segments sequentially, and then translating them into the final image), without conditioning on external inputs, and seeking to implicitly capture the conditional dependencies between objects (with the aid of the GANformer’s attention).

Indeed, **most prior GAN approaches are non-iterative**, generating the image “all-at-once” following the typical convolutional stack (from low to high resolution). These include e.g. StyleGAN and BigGAN, as well as the GANformer architecture that we have used within our model.

Consequently, **in contrast to prior iterative variational works** (DRAW, Iodine, MONet) that have been explored on either 2d or synthetic 3d images, we show that **our iterative GAN model performs successfully on various real-world domains**.

As mentioned above, we also note that our approach **incorporates additional novel components** such as the **semantic matching loss**, and that the paper provides varied experiments that explore interpretability and compositionality (responding to comments by **R2**).

---
**2. Benefits of the Recurrent Approach (*R1 question 1, R2 question 1*)**
As mentioned by **R2**, and in response to the question raised by **R1**, we observe that **the iterative sketch generation approach performs empirically better than the non-iterative counterpart**, as shown in figure 6 of the main paper and in table 1 in the supplementary (where “0” represents non-iterative generation, and 1-3 refer to the number of iterations applied for each model version).

Intuitively, we believe that the benefit in efficiency may arise from the ability of the iterative approach to potentially decompose the exponential space of possible scene layouts (which have a combinatorial number of objects arrangements) into smaller “pieces”, such that each generation step could focus on a smaller number of objects to produce rather than modeling all the scene at once. This could be especially useful for highly-structured scenes with multiple objects (such as the scene datasets we explored in the paper: COCO, CLEVR, Bedrooms and Cityscapes). The ability of the recurrent generator to condition some segments on high-resolution details of prior ones, might also help (e.g. creating a cup if the prior segment is determined to be a table).

In addition, iterative generation could allow the model to more explicitly capture phenomena of amodal completion -- first generating some of the segments (e.g. background), and then generating other overlapping segments on top of them (e.g. cars, people).  This provides us with practical means to then remove the front segments and reveal the ones behind them (see figures 4,7 for examples of that; and we will add more examples and discuss it further in the updated version of the paper!)

---
**3. Paired Layout and Image Training Data (*R3 point 3*)**
Our model relies on both layout and image data to train the two generation stages -- sketching and painting. We note in the paper that we can train the model in both the paired and the unpaired manners.
It is important for us to clarify, as also commented by **R3**, that we completely agree that the model certainly does rely on the layout data for pre-training of the sketching stage, and that this data is essential for our approach to work well, as we have discussed throughout the paper, and will also add further clarification.
The claim we made in L248 (that have been discussed by **R3**) was specifically about the ability of the model to train well when the layout and images are still available for training as two sets {layouts} and {images}, but they are simply not necessarily have to be as a set of aligned pairs of (layout_i,image_i).

In order to train the model in the unpaired manner, one can either:
1. Train both the sketching and painting stages jointly from scratch (supervising them *both* through a discriminator’s loss on the produced layouts and images),
2. Or we can first train the sketching stage on the layout dataset only, and then train the painting stage conditioned on the layouts generated by the sketching stage. In such case, both the semantic matching loss and the discriminator loss use the generated layout and image to supervise the generator).

Below, we compare the performance of the model on paired vs. unpaired generation (methods 1 and 2), showing that the model manages to perform well even in the unpaired settings, although the paired settings reach improved results. We will be adding these findings to the main paper.

---
We will post below messages to further address the questions for each of the reviews shortly. Thank you so much again for the feedback and the reviews!
-Paper 5108 authors

---

> ### Author Response · Authors · 2021-08-11
> **(continued)**
>
> **Results for Paired vs. Unpaired Data**
>
> ```
> +------------+------------+------------+------------+
> |   Dataset  |   Paired   | Unpaired 1 | Unpaired 2 |
> +------------+------------+------------+------------+
> | CLEVR      |    5.42    |    7.15    |    6.57    |
> | CelebA     |    8.59    |    9.04    |    8.95    |
> | Cityscapes |    6.21    |    7.81    |    7.32    |
> | Bedrooms   |    8.43    |    8.95    |    9.01    |
> | COCO       |   25.82    |   32.48    |   28.11    |
> +------------+------------+------------+------------+
> ```
>
> **References**
> 1. Bar, Amir, Roei Herzig, Xiaolong Wang, Gal Chechik, Trevor Darrell, and Amir Globerson. "Compositional Video Synthesis with Action Graphs."
> 2. Deng, Zhiwei, Jiacheng Chen, Yifang Fu, and Greg Mori. "Probabilistic neural programmed networks for scene generation."
> 3. Nguyen-Phuoc, Thu, Christian Richardt, Long Mai, Yong-Liang Yang, and Niloy Mitra. "BlockGAN: Learning 3d object-aware scene representations from unlabelled images."
> 4. Niemeyer, Michael, and Andreas Geiger. "GIRAFFE: Representing scenes as compositional generative neural feature fields."
> 5. Mellor, John FJ, Eunbyung Park, Yaroslav Ganin, Igor Babuschkin, Tejas Kulkarni, Dan Rosenbaum, Andy Ballard, Theophane Weber, Oriol Vinyals, and S. M. Eslami. "Unsupervised doodling and painting with improved spiral."

---

### Public Comment · ~Drew_A._Hudson1 · 2021-11-24
**Arxiv Paper**

Please see the most updated version of the paper at:
https://arxiv.org/pdf/2111.08960.pdf
Thanks!

---

### Decision · Program_Chairs · 2021-09-27

**Decision:**

Accept (Poster)

**Comment:**

- The proposed method tackles an important problem, takes a reasonable and novel approach (the loss function and recurrent generation), and the experiments are well planned, executed, and show SOTA results.
- The rebuttal addressed the major concerns from the reviewers well and thus some reviews raised the score.
- The clarity is good enough but some figures and the details of the architecture need to be clarified further.